# WNT signaling memory is required for ACTIVIN to function as a morphogen in human gastruloids

Anna Yoney[1,2], Fred Etoc[1,2], Albert Ruzo[1], Thomas Carroll[3], Jakob J Metzger[1,2], Iain Martyn[1,2], Shu Li[1], Christoph Kirst[2], Eric D Siggia[2]*, Ali H Brivanlou[1]*

[1]Laboratory of Stem Cell Biology and Molecular Embryology, The Rockefeller University, New York, United States; [2]Center for Studies in Physics and Biology, The Rockefeller University, New York, United States; [3]Bioinformatics Resource Center, The Rockefeller University, New York, United States

**Abstract** Self-organization of discrete fates in human gastruloids is mediated by a hierarchy of signaling pathways. How these pathways are integrated in time, and whether cells maintain a memory of their signaling history remains obscure. Here, we dissect the temporal integration of two key pathways, WNT and ACTIVIN, which along with BMP control gastrulation. CRISPR/Cas9-engineered live reporters of SMAD1, 2 and 4 demonstrate that in contrast to the stable signaling by SMAD1, signaling and transcriptional response by SMAD2 is transient, and while necessary for pluripotency, it is insufficient for differentiation. Pre-exposure to WNT, however, endows cells with the competence to respond to graded levels of ACTIVIN, which induces differentiation without changing SMAD2 dynamics. This cellular memory of WNT signaling is necessary for ACTIVIN morphogen activity. A re-evaluation of the evidence gathered over decades in model systems, re-enforces our conclusions and points to an evolutionarily conserved mechanism.

DOI: https://doi.org/10.7554/eLife.38279.001

**\*For correspondence:**
siggiae@rockefeller.edu (EDS);
brvnlou@rockefeller.edu (AHB)

**Competing interests:** The authors declare that no competing interests exist.

## Introduction

In the early embryo, secreted morphogens regulating a limited number of signaling pathways carry the task of instructing dynamic and coordinated cell differentiation across the developing tissue. In the day 6.5 mouse embryo, a hierarchy of signaling mediated by BMP, WNT, and NODAL carries the positional information in the epiblast leading to axis formation and germ layer specification (*Arnold and Robertson, 2009*). Each of these morphogens induces the expression of the next ligand (BMP signaling induces WNT expression, and WNT signaling induces NODAL expression), and all induce the expression of their own inhibitors. To what extent this signaling cascade is involved in human gastrulation can now be investigated using in vitro models of early human embryos derived from human embryonic stem cells (hESCs).

We have previously shown that the first step of this signaling hierarchy is conserved in humans and that in response to BMP4, hESCs grown in geometrically confined colonies, self-organize to induce and pattern embryonic and extra-embryonic germ layers. Ectoderm was specified at the center of circular colonies, extra-embryonic tissue at the edge and mesendoderm in between. Molecular signatures of gastrulation, such as the induction of SNAIL and activation of pERK could also be observed (*Warmflash et al., 2014*). Consistent with the conservation of the second step of the signaling hierarchy, BMP4, which signals through SMAD1/5/8, was shown to induce the expression of WNT ligands, which can also induce the emergence of a primitive streak and the self-organization of embryonic germ layers in this system (*Etoc et al., 2016*; *Martyn et al., 2018*). We, therefore, termed this system a human gastruloid.

**eLife digest** Embryonic stem cells can renew themselves to generate more stem cells, or specialize to become any type of cell found in an adult. They therefore hold great potential for studying how we develop from a single cell into a complex organism made of many different cell types.

In a key stage of development, individual cells form into organized tissues. The earliest phase of tissue organization involves the formation of three 'germ layers'. Human embryonic stem cells allow us to recreate this early stage of embryo development in the lab. When grown in confined spaces, the cells organize into clusters that can then develop germ layers. Previous work using these clusters showed that a network of signaling proteins – including one called WNT – trigger human embryonic stem cells to form the initial clusters. Then, another signaling protein called ACTIVIN tells the cells to specialize to form the two inner germ layers. But in experiments that apply only the ACTIVIN signal, the cells instead keep dividing to make more stem cells.

ACTIVIN can trigger the activity of a protein called SMAD. To visualize how cells respond to ACTIVIN in real time, Yoney et al. used a gene editing technique called CRISPR to add fluorescent tags to SMAD in human embryonic stem cells. The results show that the ACTIVIN response triggers a peak in the amount of SMAD in the cell's nucleus that then decreases over several hours. This briefly activated several genes that are known to help to form germ layers. However, this gene activity was not maintained for long enough to cause the stem cells to specialize and organize into layers.

Yoney et al. then repeated the experiments on cells that had previously been exposed to WNT signaling proteins. The germ layer gene activity was maintained in this case, leading to the cells specializing and forming the inner two germ layers. This suggests that the cells somehow remembered the WNT signal, and this memory changed how they responded to ACTIVIN.

The next step is to understand how cells store the memory of the WNT signal. As well as aiding our understanding of development, it could also help us to understand situations where signaling goes wrong, such as cancer. The technique used here to follow signals in real time could also be used to study other biological signaling processes.

DOI: https://doi.org/10.7554/eLife.38279.002

The third pathway, SMAD2/3, which signals on behalf of ACTIVIN and NODAL ligands, has an intriguing role. On the one hand, SMAD2/3 signaling is required for hESC pluripotency maintenance (*James et al., 2005*; *Vallier et al., 2005*). On the other hand, the ACTIVIN/NODAL pathway is necessary for fate specification, acting as a morphogen to pattern the blastula of vertebrate embryos from the amphibian to the mouse (*McDowell and Gurdon, 1999*; *Robertson, 2014*). The ACTIVIN/NODAL pathway is also necessary for the self-organization of human gastruloids as inhibition of SMAD2/3 signaling blocks primitive streak formation and mesendoderm induction by BMP4 and eliminates anterior mesendoderm fates induced by WNT3A (*Warmflash et al., 2014*; *Martyn et al., 2018*). Furthermore, co-presentation of ACTIVIN with WNT3A to micropatterned hESC colonies leads to the induction of the organizer specific marker GOOSECOID, and a functional human primitive streak, which when grafted into chick embryos induces the formation of a secondary axis (*Martyn et al., 2018*).

Despite these findings, the mechanism by which SMAD2/3 signaling can specify both hESC pluripotency and differentiation remains perplexing, and a number of key questions remain unanswered. Among them are: how can a single signaling pathway carry these two opposite functions before and after the onset of gastrulation? To what extent do the dynamics of SMAD signaling affect these readouts? Finally, to what extent do cells have a memory of past signaling?

In this study we aimed to provide highly quantitative answers to these questions by following the dynamics of TGFβ signaling during the self-organization of human gastruloids. Imaging of hESC reporter lines engineered by CRISPR/Cas9-mediated tagging of SMAD1, SMAD2, or SMAD4 demonstrated that each branch of the pathway has distinct signaling dynamics. In response to ACTIVIN, SMAD2 displayed a dramatic transient nuclear translocation, which stood in sharp contrast to the stable BMP4-induced SMAD1 response. ACTIVIN stimulation did induce transient mesendodermal

gene transcription, which correlated with SMAD2 dynamics. This induction, however, was not sustained and cells reverted back to pluripotency at later times. Interestingly, pre-presentation of WNT3A to the cells, while not changing SMAD2 dynamics or expression of the pluripotency markers, stabilized the subsequent transcriptional response to ACTIVIN to maintain mesendodermal fates. This implies an unexpected ability of human embryonic stem cells to record their signaling history without overt changes in fate.

## Results

### Gastruloids respond to ACTIVIN stimulation at the colony border

A functional SMAD2/3 pathway is necessary for both maintenance of pluripotency, as well as differentiation and self-organization of human gastruloids downstream of BMP4 and WNT3A (*Warmflash et al., 2014*; *Martyn et al., 2018*). In the context of BMP4 induced differentiation, treatment with a small molecule inhibitor of SMAD2/3 signaling, SB431542 (SB), eliminated all mesendodermal fates, as indicated by the loss of BRA and SOX17 positive cells, leaving only the putative extra-embryonic and ectoderm fates, marked by CDX2 and SOX2 expression, respectively (*Figure 1A*) (*Warmflash et al., 2014*). Presentation of the downstream morphogen WNT3A induced primitive streak formation at the colony border, and addition of SB in this context removed the SOX17 positive mesendodermal population (*Figure 1A*) (*Martyn et al., 2018*). In order to ask whether gastruloid self-organization can be initiated at the ACTIVIN/NODAL point in the signaling hierarchy, we stimulated micropatterned colonies grown in conditioned media with high concentrations of ACTIVIN A (referred to as ACTIVIN throughout this work). In contrast to stimulation with BMP4 and WNT3A, no differentiation was observed after 48 hr with ACTIVIN alone (*Figure 1B*). Surprisingly, the lack of differentiation in response to ACTIVIN was not due to a lack of signal sensing, as an increase in nuclear SMAD2/3 was detected by immunofluorescence at the colony border after one hour of stimulation (*Figure 1C–D*). We conclude that while SMAD2/3 signaling is necessary for mesendoderm induction downstream of BMP4 and WNT3A in human gastruloids, it is not sufficient to induce it.

### Two branches of the TGFβ pathway display different signaling dynamics

The inability of ACTIVIN/SMAD2 to induce differentiation stood in stark contrast with the ability of BMP/SMAD1 to induce gastruloid self-organization, including patterning of the mesoderm and endoderm germ layers. This contrast in activity of the two branches of the TGFβ pathway prompted us to assess possible differences in their signaling dynamics. We used CRISPR/Cas9 genome engineering on RUES2 to fluorescently tag the N-terminus of the endogenous receptor-associated, R-SMAD1, with tagRFP (RUES2-RFP-SMAD1) and R-SMAD2 with mCitrine (RUES2-mCit-SMAD2) (*Figure 2A* and *Figure 2—figure supplement 1*). As activation of the pathway leads to the binding of SMAD1 and 2 to the co-SMAD, SMAD4, before nuclear translocation and regulation of gene expression, a GFP-tagged SMAD4 line (RUES2-GFP-SMAD4) was also included (*Figure 2A*) (*Nemashkalo et al., 2017*). Each line was also transfected with ePiggyBac transposable elements carrying a nuclear marker (H2B-mCitrine or H2B-mCherry) in order to analyze the response of individual cells (*Figure 2—figure supplement 2A*). N-terminal SMAD fusion proteins were shown to function similarly to endogenous proteins in biochemical and cell-based assays (*Schmierer and Hill, 2005*). Additionally, the SMAD response dynamics measured with our reporter lines, matched the behavior by of the endogenous proteins measured by immunofluorescence and western blotting as discussed below.

We began our dynamic studies of the two branches of TGFβ signaling in RUES2 cells grown on micropatterned colonies in chemically-defined medium (TeSR-E7), which is a version of serum-free E8 medium that can maintain hESCs and that lacks any TGFβ ligands (*Chen et al., 2011*). Therefore, in E7 the exogenous TGFβ levels could be precisely controlled. In response to BMP4, we detected an increase in the SMAD1 nuclear signal that was stable over 12 hr (*Figure 2B–D*, *Figure 2—figure supplement 2B*, *Figure 2—videos 1* and *2*). The SMAD1 response was observed at the colony edge, consistent with our previous immunofluorescence results and our discovery that the TGFβ receptors are localized to the apical surface only at the border of the colony (*Warmflash et al., 2014*; *Etoc et al., 2016*). In our previous work we quantified the SMAD1 response detected by

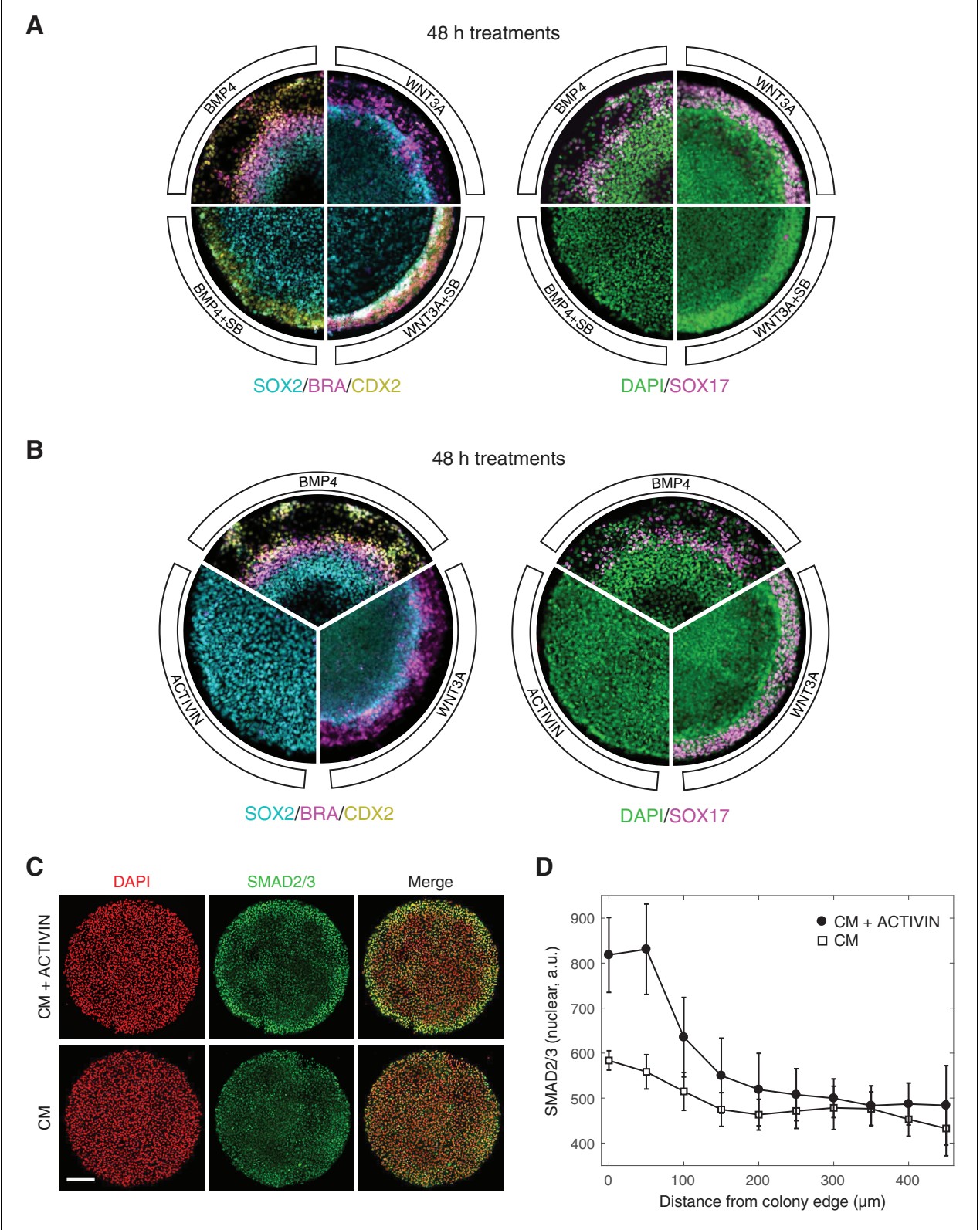

**Figure 1.** ACTIVIN modifies gastruloid differentiation but it cannot induce it. (**A**) Micropatterned colonies in conditioned media treated for with BMP4 (50 ng/mL), BMP4 (50 ng/mL) +SB (10 μM), WNT3A (100 ng/mL), or WNT3A (100 ng/mL) +SB (10 μM) for 48 hr. The colonies were fixed and analyzed by immunofluorescence. Left: SOX2 (cyan), BRA (magenta), CDX2 (yellow). Right: DAPI (green), SOX17 (magenta). (**B**) Micropatterned colonies in conditioned media treated with BMP4 (50 ng/mL), WNT3A (100 ng/mL), or ACTIVIN (100 ng/mL) for 48 hr. The colonies were fixed and analyzed by

*Figure 1 continued on next page*

*Figure 1 continued*

immunofluorescence. Left: SOX2 (cyan), BRA (magenta), CDX2 (yellow). Right: DAPI (green), SOX17 (magenta). (**C–D**) Micropatterned colonies in conditioned media (CM) treated with ACTIVIN (100 ng/mL) for 1 hr or left untreated. The colonies were fixed and analyzed by immunofluorescence. (**C**) Images: DAPI (red), SMAD2/3 (green). Scale bar, 200 μm. (**D**) Quantification of the mean SMAD2/3 nuclear fluorescence as a function of radial position from the colony edge: CM + ACTIVIN (filled circles), CM (open squares). Error bars represent the standard deviation across n = 5 (CM) and n = 6 (CM + ACTIVIN) colonies from one experiment. All micropatterned culture experiments were performed on at least three separate occasions with similar results.

DOI: https://doi.org/10.7554/eLife.38279.003

immunofluorescence as the number of positive nuclei within the colony (*Etoc et al., 2016*). Although our reporter line allows us to track the level of nuclear SMAD1 in individual cells, as opposed to a binary on/off quantification, we obtain similar results to those shown in *Figure 2D* with a binary analysis of the RFP-SMAD1 reporter line (*Figure 2—figure supplement 2C–D*). In response to ACTIVIN, SMAD2 on the other hand, responded only transiently: a pulse of nuclear translocation over the first 1–2 hr was followed by a gradual decrease over the next 6 hr (*Figure 2E–G*, *Figure 2—figure supplement 2E*, and *Figure 2—video 3*). Interestingly, the long-term SMAD2 nuclear level did not return completely to the pre-stimulus level. As in the case of SMAD1, the SMAD2 response was highest at the colony edge, again consistent with the apical localization of the receptors (*Etoc et al., 2016*).

## Single-cell response dynamics reflect the behavior at the edge of gastruloids

In order to study signaling dynamics at the single-cell level and to eliminate modifier influences on both SMAD branches coming from neighboring cells within the micropatterned colony, we performed the same experiment on dissociated cells grown under regular culture conditions (*Figure 3A*). The SMAD1 response to BMP was stable and the level of nuclear RFP-SMAD1 was dependent on the BMP4 ligand concentration in agreement with previous single-cell immunofluorescence measurements (*Figure 3B–C*) (*Etoc et al., 2016*). The increase in the average nuclear signal as a function of time resulted from nearly all cells responding to the added BMP4, which is evident in the shift in the histogram of RFP-SMAD1 nuclear intensity (*Figure 3—figure supplement 1A*). The SMAD2 response to ACTIVIN was again transient with a nuclear signal that remained elevated at longer times as demonstrated by our reporter line and by immunofluorescence and western blot analysis of the parental RUES2 line (*Figure 3D–E*, *Figure 3—figure supplement 1B–C*). This is consistent with other data based on immunofluorescence (*Heemskerk et al., 2017*). Although, the adaptive response is somewhat more pronounced in our experimental setup. As with the SMAD1 response, the average SMAD2 response resulted from nearly all cells responding with similar dynamics, which is evident in the shift up and down in the histogram of SMAD2 nuclear intensity (*Figure 3—figure supplement 1D*). The SMAD2 peak response displayed a strong sigmoidal dependence on ACTIVIN concentration (*Figure 3—figure supplement 1E*). However, the post-stimulation baseline, defined as the average SMAD2 nuclear-to-cytoplasmic ratio at T > 8 hr after ACTIVIN addition, as well as the time scale of the transient response, was not dependent on the ACTIVIN dose above 0.5 ng/mL (*Figure 3—figure supplement 1E–F*). This transient SMAD2 signaling behavior did not result from depletion of ACTIVIN from the medium, as culture medium recovered from cells that were incubated with ACTIVIN for 12 hr, still induced a SMAD2 response when presented to unstimulated cells (*Figure 3—figure supplement 1G*). SMAD4 followed the dynamics of the relevant R-SMADs following presentation of BMP4 or ACTIVIN (*Figure 3F–G*, *Figure 3—figure supplement 1H*). The stable and adaptive responses of SMAD4 to BMP and ACTIVIN, respectively, is consistent with recent experiments that also utilized this reporter line (*Nemashkalo et al., 2017*; *Heemskerk et al., 2017*). Thus, in response to their activating ligands, the two branches of the TGFβ pathway display distinct dynamics of signal transduction for both the R-SMAD and the co-SMAD. This also demonstrates that modifying signals do not influence the dynamics of the response at the edge of the micropatterned colonies.

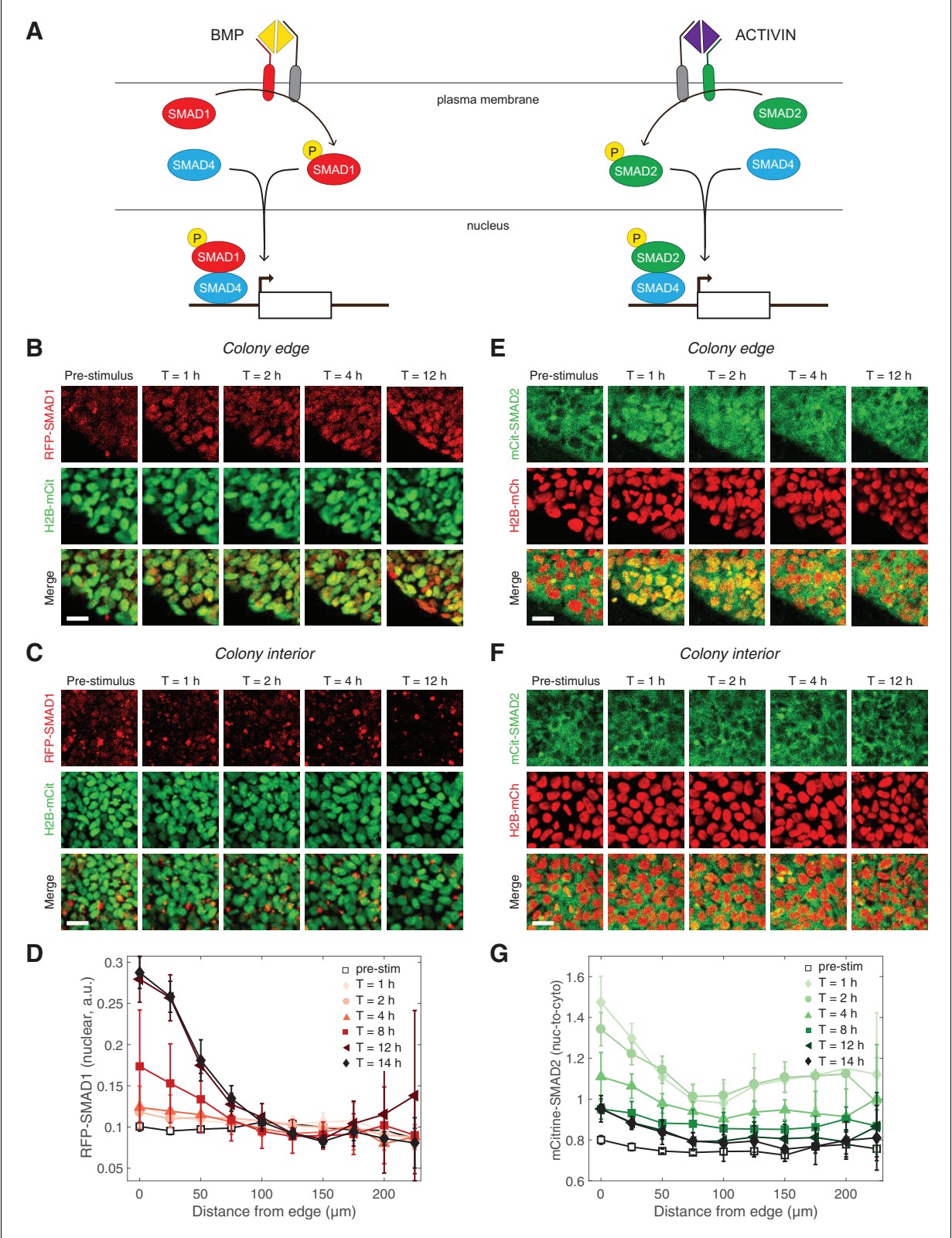

**Figure 2.** Two branches of the TGFβ pathway display different signaling dynamics. (**A**) BMP and ACTIVIN signaling represent the two branches of the TGFβ pathway. BMP signals through R-SMAD1 and ACTIVIN signals through R-SMAD2. R-SMADs form a complex with co-SMAD4 and the complex regulates target gene expression. (**B–C**) RUES2-RFP-SMAD1 grown on micropatterned colonies in E7 and stimulated with BMP4 (50 ng/mL). The response of cells at the colony edge (**B**) and near the colony center (**C**) as a function of time following BMP4 presentation. The intensity range was

*Figure 2 continued on next page*

*Figure 2 continued*
adjusted to the same minimum and maximum values in all images in both (**B**) and (**C**). Scale bars, 25 μm. (**D**) Average RFP-SMAD1 nuclear signal as a function of radial position within the colony at different time points following BMP4 treatment. The single-cell nuclear RFP-SMAD1 intensity was quantified and normalized to the single-cell H2B-mCitrine signal. Error bars represent the standard deviation over n = 5 colonies from one experiment. (**E–F**) RUES2-mCit-SMAD2 grown on micropatterned colonies in E7 and stimulated with ACTIVIN (10 ng/mL). The response of cells at the colony edge (**E**) and near the colony center (**F**) as a function of time following ACTIVIN presentation. The intensity range was adjusted to the same minimum and maximum values in all images in both (**E**) and (**F**). Scale bars represent 25 μm. (**G**) Average mCitrine-SMAD2 nuclear-to-cytoplasmic signal as a function of radial position within the colony at different time points following ACTIVIN treatment. The single-cell nuclear mCitrine intensity was quantified and normalized to the single-cell cytoplasmic mCitrine signal. Error bars represent the standard deviation over n = 5 colonies from one experiment.
DOI: https://doi.org/10.7554/eLife.38279.004
The following video and figure supplements are available for figure 2:

**Figure supplement 1.** R-SMAD reporter line generation.
DOI: https://doi.org/10.7554/eLife.38279.005
**Figure supplement 2.** Quantification of R-SMAD dynamics at the single-cell level in micropatterned colonies.
DOI: https://doi.org/10.7554/eLife.38279.006
**Figure 2—video 1.** RUES2-RFP-SMAD1 micropatterned colony stimulated with BMP4.
DOI: https://doi.org/10.7554/eLife.38279.007
**Figure 2—video 2.** RUES2-RFP-SMAD1 micropatterned colony unstimulated.
DOI: https://doi.org/10.7554/eLife.38279.008
**Figure 2—video 3.** RUES2-mCit-SMAD2 micropatterned colony stimulated with ACTIVIN.
DOI: https://doi.org/10.7554/eLife.38279.009

## ACTIVIN elicits a transient and stable transcriptional response

We have previously shown that BMP4 signaling induces a sustained transcriptional response leading to gastruloid differentiation (*Warmflash et al., 2014*; *Etoc et al., 2016*). This is consistent with the stable nature of SMAD1 signaling presented above. The adaptive behavior of SMAD2 signaling prompted us to ask whether the short SMAD2 signaling peak was sufficient to elicit a transcriptional response and fate changes in RUES2 cells exposed to ACTIVIN. RNA-seq analysis was performed on dissociated cells cultured in E7 and E7 +ACTIVIN at 1, 2.5, 4, 8 and 12 hr following stimulation. 3529 genes showed a change in expression level of at least two-fold during the experimental time course. They fell into three distinct groups. The first, which consisted of the majority of transcripts (2,956), peaked at 2.5 hr and declined at later time points (*Figure 4A*, magenta box). This group matched the timing of the transient SMAD2 response and it included key regulators of mesendodermal differentiation, such as EOMES, HHEX, GATA2, and GATA3 (*Figure 4—source data 1*) (*Teo et al., 2011*; *Loh et al., 2014*). The second group, which consisted of 452 transcripts, showed stable induction (*Figure 4A*, orange box). This group included genes expressed during pluripotency, such as NANOG, NODAL, LEFTY1, LEFTY2 and SMAD7 (*Figure 4—source data 2*) (*Sato et al., 2003*). Finally, the third group, which consisted of 121 transcripts, represented genes that were stably or transiently down regulated upon ACTIVIN presentation and included genes that are involved in signaling pathways not previously associated with pluripotency or differentiation, such as insulin signaling and cAMP response (*Figure 4A*, gray box and *Figure 4—source data 3*). These results suggest that cells transiently activate differentiation in response to ACTIVIN.

Examination of the signaling hierarchy involved in gastruloid self-organization revealed the presence of feedback loops at all three levels. ACTIVIN induced the expression of its own ligands and inhibitors, as well as those of the BMP and WNT pathway (*Figure 4—source data 1–2*). However, despite the induction of the expression of the ligands and inhibitors, the overall threshold of signaling was not sufficient to induce and maintain mesendodermal fates from either the BMP or the WNT pathway.

We took two additional approaches to evaluate the transient and stable transcriptional responses following ACTIVIN treatment. First, we performed motif enrichment analysis on our gene groups. We selected 10 transcription factors that regulate primitive streak and mesendodermal differentiation: MIXL1, LEF1, BRACHYURY (BRA), GATA6, FOXH1, FOXA1, FOXA2, GOOSECOID (GSC), SOX17, and EOMES, and asked if their binding sites are enriched in the promoter region of the genes belonging to each of the dynamic groups. In support of our hypothesis, the motifs were significantly enriched only in the transiently expressed genes of group one and not within group 2 or 3

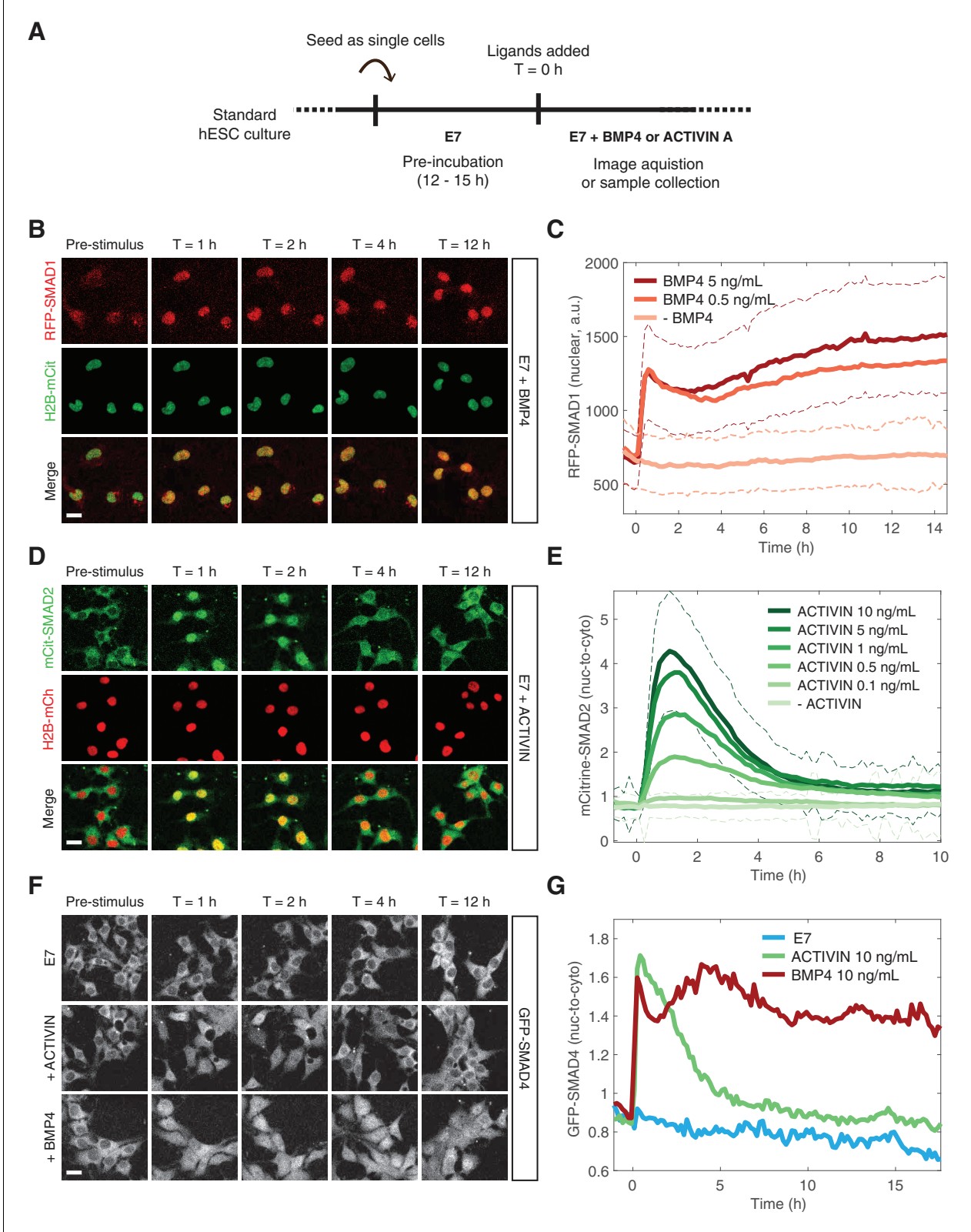

**Figure 3.** BMP and ACTIVIN elicit different single-cell SMAD response dynamics. (**A**) Schematic outlining the single-cell experimental protocol. (**B**) RFP-SMAD1 response in single cells to BMP4 (5 ng/mL, added at T = 0 hr). (**C**) Quantification of the RFP-SMAD1 nuclear signal as a function of time and BMP4 concentration. Images were acquired every 10 min. Solid lines represent the average response at each time point (n > 200 cells per time point). Dashed lines represent the population standard deviation for – BMP4 (light red) and BMP4 (5 ng/mL, dark red) conditions. Similar results were obtained

*Figure 3 continued on next page*

*Figure 3 continued*

in two independent experiments. (D) mCitrine-SMAD2 response in single cells to ACTIVIN (10 ng/mL, added at T = 0 hr). (E) Quantification of the mCitrine-SMAD2 nuclear-to-cytoplasmic ratio as a function of time and ACTIVIN concentration. Images were acquired every 10 min. Solid lines represent the average response at each time point (n > 200 cells per time point). Dashed lines represent the population standard deviation for – ACTIVIN (lightest green) and ACTIVIN (10 ng/mL, darkest green) conditions. Data were collectively obtained from three independent experiments. (F) GFP-SMAD4 response in single cells to ACTIVIN (10 ng/mL) or BMP4 (10 ng/mL) or cells that were left untreated (E7). Ligands were added at T = 0 hr. (G) Quantification of the GFP-SMAD4 nuclear-to-cytoplasmic ratio as a function of time in E7 (blue), E7 +BMP4 (10 ng/mL, red), or E7 +ACTIVIN (10 ng/ mL, green). Images were acquired every 10 min. Solid lines represent the average response at each time point (n > 200 cells per time point). Similar results were obtained in two independent experiments. Scale bars, 25 μm.

DOI: https://doi.org/10.7554/eLife.38279.010

The following figure supplement is available for figure 3:

**Figure supplement 1.** BMP and ACTIVIN elicit different single-cell SMAD response dynamics.

DOI: https://doi.org/10.7554/eLife.38279.011

(*Figure 4—figure supplement 1A* and *Figure 4—source data 4*). This suggests that the gene regulatory network activated during the peak of SMAD2 signaling is associated with mesendodermal differentiation. We additionally compared our gene groups with tissue specific genes identified in isolated endoderm, mesoderm, and ectoderm/epiblast tissue from E7.5 mouse embryos (*Lu et al., 2018*). Although all groups contained some significant enrichment of genes from one or more of the mouse germ layers, group one displayed the most significant enrichment of endodermal genes (*Figure 4—figure supplement 1B* and *Figure 4—source data 5*). Overall our data demonstrate that during the peak of SMAD2 nuclear accumulation, hESCs are *en route* for differentiation. However, the mesendodermal differentiation program is not maintained and cells return to pluripotency.

## SMAD3 is dispensable for the response of the adaptive and stable gene classes to ACTIVIN

In the mouse embryo, SMAD3 is dispensable for early development as demonstrated by the fact that SMAD3 knockout mice make it to adulthood (*Zhu et al., 1998*; *Yang et al., 1999*; *Datto et al., 1999*). In the absence of SMAD2 in the epiblast, SMAD3 can mediate some mesoderm induction during gastrulation. However, more anterior mesendodermal lineages are completely eliminated and the embryos fail at gastrulation suggesting a critical role for SMAD2 in this process (*Vincent et al., 2003*; *Dunn et al., 2004*). In order to decipher whether the transient SMAD2 response is sufficient to drive the transcriptional program downstream of ACTIVIN presentation in hESCs, we generated two independent RUES2 SMAD3 knockout lines (RUES2-SMAD3$^{-/-}$) using CRISPR-Cas9 mediated genome editing (*Figure 4—figure supplement 2A–B*). These lines maintained expression of pluripotency markers NANOG, OCT4, and SOX2, which is consistent with the previous finding that SMAD2, but not SMAD3, regulates NANOG expression to promote pluripotency in hESCs and mouse epiblast stem cells (*Figure 4—figure supplement 1C*) (*Sakaki-Yumoto et al., 2013*). In response to ACTIVIN, RUES2-SMAD3$^{-/-}$ cells displayed a transcriptional response identical to the parental RUES2 line for both pluripotency- and mesendoderm-associated ACTIVIN target genes (*Figure 4B–C*). Although our results cannot rule out possible redundancy between SMAD2 and SMAD3, we conclude that SMAD3 is not required for maintenance of pluripotency or the transcriptional response dynamics following ACTIVIN presentation.

## Long-term, elevated SMAD2 baseline maintains pluripotency

We have shown that SMAD2 nuclear levels are transient following a step input of ACTIVIN, with a long-term baseline that remains elevated relative to the pre-stimulus level. We next asked if the increase in the SMAD2 baseline after the peak response is required for the maintenance of pluripotency. In order to address this question, RUES2-mCit-SMAD2 cells were treated with SB in order to inhibit ACTIVIN signaling, 8 hr after stimulation, and analyzed for their ability to maintain pluripotency. SB treatment led to a decrease in SMAD2 nuclear-to-cytoplasmic levels back to the unstimulated baseline (*Figure 5A*). As observed previously, presentation of SB led to a loss of pluripotency, as indicated by the loss of NANOG expression in RUES2-mCit-SMAD2 (*Figure 5B*) (*James et al., 2005*; *Vallier et al., 2005*; *Xu et al., 2008*). Analysis of the parental RUES2 line confirmed the loss of pluripotency under the same experimental conditions (*Figure 5—figure supplement 1A*). We

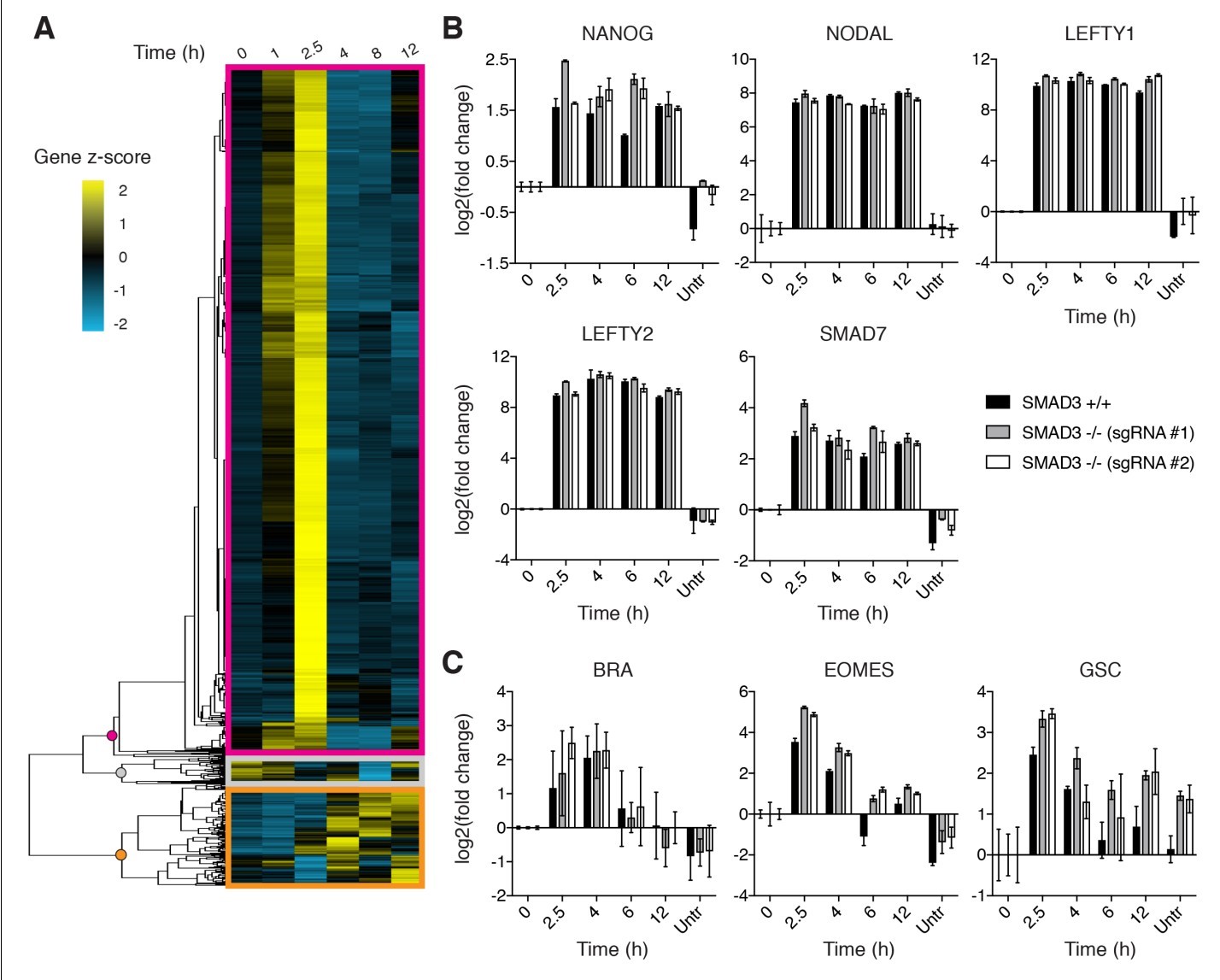

**Figure 4.** ACTIVIN/SMAD2 signaling results in stable and transient transcriptional responses. (A) Hierarchical clustering of RNA-seq time course data for genes showing a fold-change >2 in response to ACTIVIN (10 ng/mL) relative to the pre-stimulus level (T = 0 hr). The gene z-score at each time point was calculated by subtracting the average and dividing by the standard deviation of the normalized read counts across all time points. (B–C) RT-PCR analysis of (B) pluripotency- and (C) mesendoderm-associated genes following presentation of ACTIVIN (10 ng/mL) in the parental RUES2 line (black bars) and the RUES2-SMAD3$^{-/-}$ lines (gray and white bars). An additional sample was collected that was left untreated for the 12 hr time course (untr). Expression in each sample was normalized to GAPDH and then to the pre-stimulus level (T = 0 hr). Data represents the mean ±S.D. for n = 3 technical replicates. Similar results were obtained in three independent experiments using the parental RUES2 line.
DOI: https://doi.org/10.7554/eLife.38279.012

The following source data and figure supplements are available for figure 4:

**Source data 1.** Hierarchical clustering group one genes (z-score).
DOI: https://doi.org/10.7554/eLife.38279.015
**Source data 2.** Hierarchical clustering group two genes (z-score).
DOI: https://doi.org/10.7554/eLife.38279.016
**Source data 3.** Hierarchical clustering group three genes (z-score).
DOI: https://doi.org/10.7554/eLife.38279.017
**Source data 4.** Motif enrichment analysis for gene groups 1–3.
DOI: https://doi.org/10.7554/eLife.38279.018
**Source data 5.** Gene enrichment analysis for gene groups 1–3.
*Figure 4 continued on next page*

*Figure 4 continued*

DOI: https://doi.org/10.7554/eLife.38279.019

**Source data 6.** Overlapping genes obtained from gene enrichment analysis.

DOI: https://doi.org/10.7554/eLife.38279.020

**Figure supplement 1.** Motif and gene enrichment analysis for ACTIVIN-responsive gene groups.

DOI: https://doi.org/10.7554/eLife.38279.013

**Figure supplement 2.** RUES2-SMAD3$^{-/-}$ line generation.

DOI: https://doi.org/10.7554/eLife.38279.014

conclude that the elevated baseline at the tail of the SMAD2 response is ligand dependent and responsible for maintaining the pluripotency program long-term.

The fact that the SMAD2 post-stimulation baseline is the same regardless of ACTIVIN concentration above 0.5 ng/mL, suggests that pluripotency is insensitive to graded ligand levels above this threshold (*Figure 3E*). To test this hypothesis, we treated single cells with different levels of ACTIVIN and compared the expression of NANOG, OCT4, and SOX2 after 2 days of stimulation. Expression of all three markers was similar at three different concentrations of added ACTIVIN (1, 10, and 100 ng/mL) and expression of NANOG and OCT4 was elevated relative to the –ACTIVIN condition (*Figure 5C–D* and *Figure 5—figure supplement 1B*; results from additional hESC line RUES1 shown in *Figure 5—figure supplement 1C*). To test for zero ACTIVIN/SMAD2 input we presented SB for the same amount of time. Pluripotency was not maintained under these conditions as demonstrated by down-regulation of NANOG and OCT4 below the –ACTIVIN condition (*Figure 5C–D* and *Figure 5—figure supplement 1B–C*). Toxicity as a result of SB treatment or increased ACTIVIN levels was not observed (*Figure 5—figure supplement 1D*). This confirms the insensitivity of the pluripotent state to graded ACTIVIN. Even at the highest ligand concentrations pluripotency was maintained and no differentiation was observed.

In order to compare immunofluorescence data across biological replicates, which may differ in absolute intensity values, we computed the Kolmogorov-Smirnov (KS) distance between each marker's cumulative distribution function (CDF) measured under different conditions to a reference CDF, in this case the CDF measured under the 1 ng/mL ACTIVIN condition (*Figure 5—figure supplement 1E*). In biological replicates using RUES2 or RUES1 the distributions of NANOG and OCT levels under the 100 or 10 ng/mL ACTIVIN conditions are reproducibly close to the distributions measured under the 1 ng/mL ACTIVIN condition (small KS distance, *Figure 5E*). Whereas the distributions measured under the –ACTIVIN and SB conditions are reproducibly far from the distributions measured under the 1 ng/mL ACTIVIN condition (large KS distance, *Figure 5E*). This analysis demonstrates the reproducibility of our findings that the pluripotent state is insensitive to graded ACTIVIN levels.

## WNT priming unveils ACTIVIN-dependent mesendoderm differentiation

We have demonstrated that ACTIVIN alone cannot drive stable differentiation of human gastruloids. However, WNT can lead to differentiation and self-organization of gastruloids in a SMAD2/3 dependent manner (*Martyn et al., 2018*). Since WNT is operating up-stream of ACTIVIN/NODAL in the proposed signaling hierarchy, we asked whether cells with a history of WNT signaling might respond differently to ACTIVIN treatment. To address this, cells were treated with WNT for 24 hr, washed to remove WNT, and then cultured with or without ACTIVIN for an additional 24 hr (*Figure 6A*). Surprisingly, when cells were treated with WNT alone in the absence of ACTIVIN no mesendodermal differentiation was observed and cells remained pluripotent (*Figure 6B–C*). However, if the cells were exposed to WNT before ACTIVIN stimulation mesendodermal fates were robustly induced (*Figure 6D–E*). KS distance analysis using the pluripotency condition (-/ACT) as the reference condition demonstrated the reproducibility of these observations in RUES2 and RUES1 (*Figure 6F–G*). Following WNT and ACTIVIN treatment, mesendoderm marker expression was also observed in the RUES2-SMAD3$^{-/-}$ line, suggesting that SMAD3 is not necessary for ACTIVIN-dependent mesendoderm induction (*Figure 6—figure supplement 1A*). Differentiation was ACTIVIN concentration dependent, as demonstrated by the induction of BRA at low ACTIVIN and the induction of BRA, EOMES and GSC at high ACTIVIN concentrations (*Figure 6—figure supplement 1B–D*). Therefore,

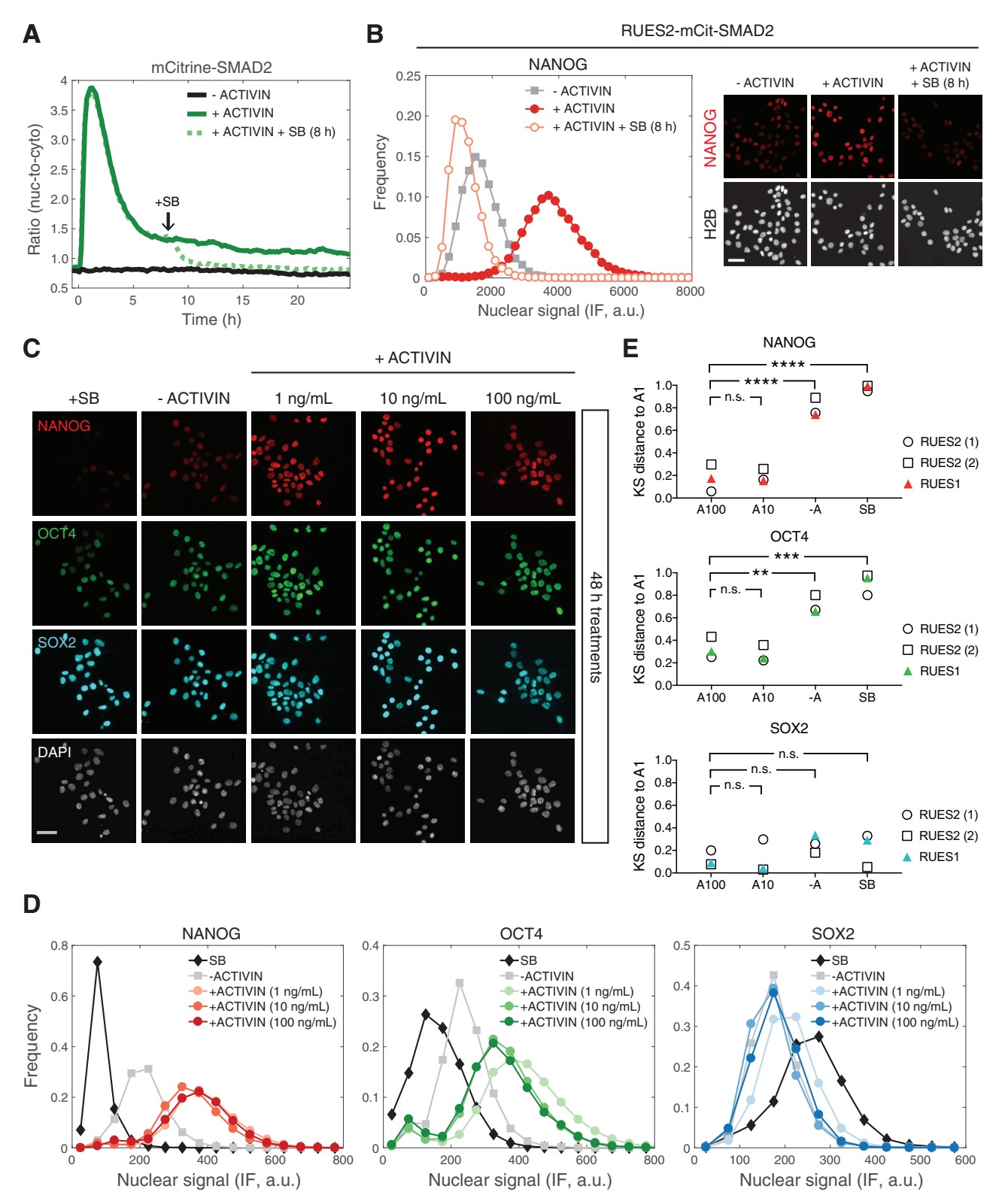

**Figure 5.** Long-term, elevated SMAD2 baseline maintains pluripotency. (A) Quantification of the mCitrine-SMAD2 nuclear-to-cytoplasmic ratio following treatment with ACTIVIN (10 ng/mL added at T = 0 hr, solid green line). Following the transient SMAD2 response, SB (10 μM) was added to one of the samples (dotted green line, added at T = 8 hr). A third sample was left untreated in E7 (–ACTIVIN) for the duration of the experiment (solid black line). Images were acquired every 10 min. Lines represent the average response at each time point (n > 200 cells per time point). Similar results

*Figure 5 continued on next page*

*Figure 5 continued*

were obtained in two independent experiments. (B) The samples in A were fixed 24 hr after ACTIVIN addition and analyzed for NANOG expression by immunofluorescence (IF). Histograms show the nuclear IF signal quantified in single cells (n > 5,000 cells per condition). Scale bar, 50 μM. (C–D) RUES2 cells were cultured in E7 with different levels of Activin (0, 1, 10, 100 ng/mL) or SB (10 μM) for 2 days. Cells were fixed and analyzed by immunofluorescence (IF). (C) Images: NANOG (red), OCT4 (green), SOX2 (cyan), DAPI (gray). Scale bar, 50 μM. (D) Histograms showing the nuclear IF signal quantified in single cells (n > 5,000 cells per condition). Similar results to those shown in C and D were obtained in two independent experiments in RUES2 and in an additional hESC line, RUES1 (see *Figure 5—figure supplement 1B–C*). (E) Kolmogorov-Smirnov (KS) distance of the cumulative probability distribution (CDF) of each marker to the reference CDF (1 ng/mL ACTIVIN condition) for independent experiments in RUES2 and in RUES1. n.s., not significant, **p<0.01, ***p<0.001, ****p<0.0001, ANOVA.
DOI: https://doi.org/10.7554/eLife.38279.021

The following figure supplement is available for figure 5:

**Figure supplement 1.** Long-term, elevated SMAD2 baseline maintains pluripotency.
DOI: https://doi.org/10.7554/eLife.38279.022

following WNT priming ACTIVIN functions as a morphogen to pattern mesendodermal fates and SMAD3 is again dispensable for this process.

## WNT memory stabilizes the mesendodermal transcriptional response to ACTIVIN

In order to address the mechanism of WNT priming, we first asked whether the transcriptional effector of canonical WNT signaling, β-catenin, is involved in this process. We therefore treated cells with the small molecule inhibitor endo-IWR-1, which blocks β-catenin function through stabilization of its destruction complex. endo-IWR-1 was added with WNT on day one and again with ACTIVIN on day 2. In the presence of endo-IWR-1 mesendoderm maker expression was eliminated and pluripotency marker expression was maintained, indicating a requirement for β-catenin in the differentiation process (*Figure 7A* and *Figure 7—figure supplement 1A*). We then asked if endogenous WNT, induced by a possible positive feedback loop, is required for stable induction of mesendoderm, particularly during the ACTIVIN treatment phase when exogenous WNT has been removed. To address this we blocked endogenous WNT secretion using the small molecule inhibitor IWP-2, which was added with WNT on day one and again with ACTIVIN on day 2. We find that addition of IWP-2 does not affect ACTIVIN-dependent mesendoderm differentiation and loss of pluripotency (*Figure 7A* and *Figure 7—figure supplement 1A*). However, IWP-2 at the same concentration does block WNT-dependent mesendoderm differentiation downstream of BMP4 (*Figure 7—figure supplement 1B*) (*Martyn et al., 2018*). Toxicity as a result of either endo-IWR-1 or IWP-2 treatment was not observed (*Figure 7B*). Together, these results demonstrate the presence of an unexpected WNT signaling memory in cells that is mediated via β-catenin and is established prior to and is required for the morphogen activity of ACTIVIN.

Since we have shown that the SMAD response dynamics can be stable or transient depending on the branch being activated, we next asked if WNT memory affects the dynamics of SMAD2 signal transduction. The transient response of SMAD2 and SMAD4 and the elevation in the baseline post-stimulation was the same whether or not cells were previously exposed to WNT (*Figure 7C*). However, transcription of mesendodermal genes was stabilized in response to ACTIVIN following WNT priming (*Figure 7D*). As in the case of pluripotency, stable mesendodermal fate acquisition required on-going SMAD2 signaling in the elevated baseline as treatment with SB, 8 hr after ACTIVIN stimulation, eliminated mesendodermal differentiation at 24 hr after ACTIVIN (*Figure 7—figure supplement 1D*). In contrast, and further arguing for a mechanism of WNT priming that is temporally upstream of ACTIVIN, addition of SB during WNT priming did not block mesendoderm differentiation (*Figure 7—figure supplement 1E*). Overall, we conclude that cells maintain a memory of WNT exposure, which makes them competent to differentiate in response to ACTIVIN (*Figure 7E*).

## Discussion

Following the tradition of experimental embryology, during the late 1960 s, Nieuwkoop began a series of explant and transplant experiments in the amphibian blastula that ultimately led to the discovery of ACTIVIN as a mesoderm inducer in the 1980 s (*Nieuwkoop, 1969*; *McDowell and*

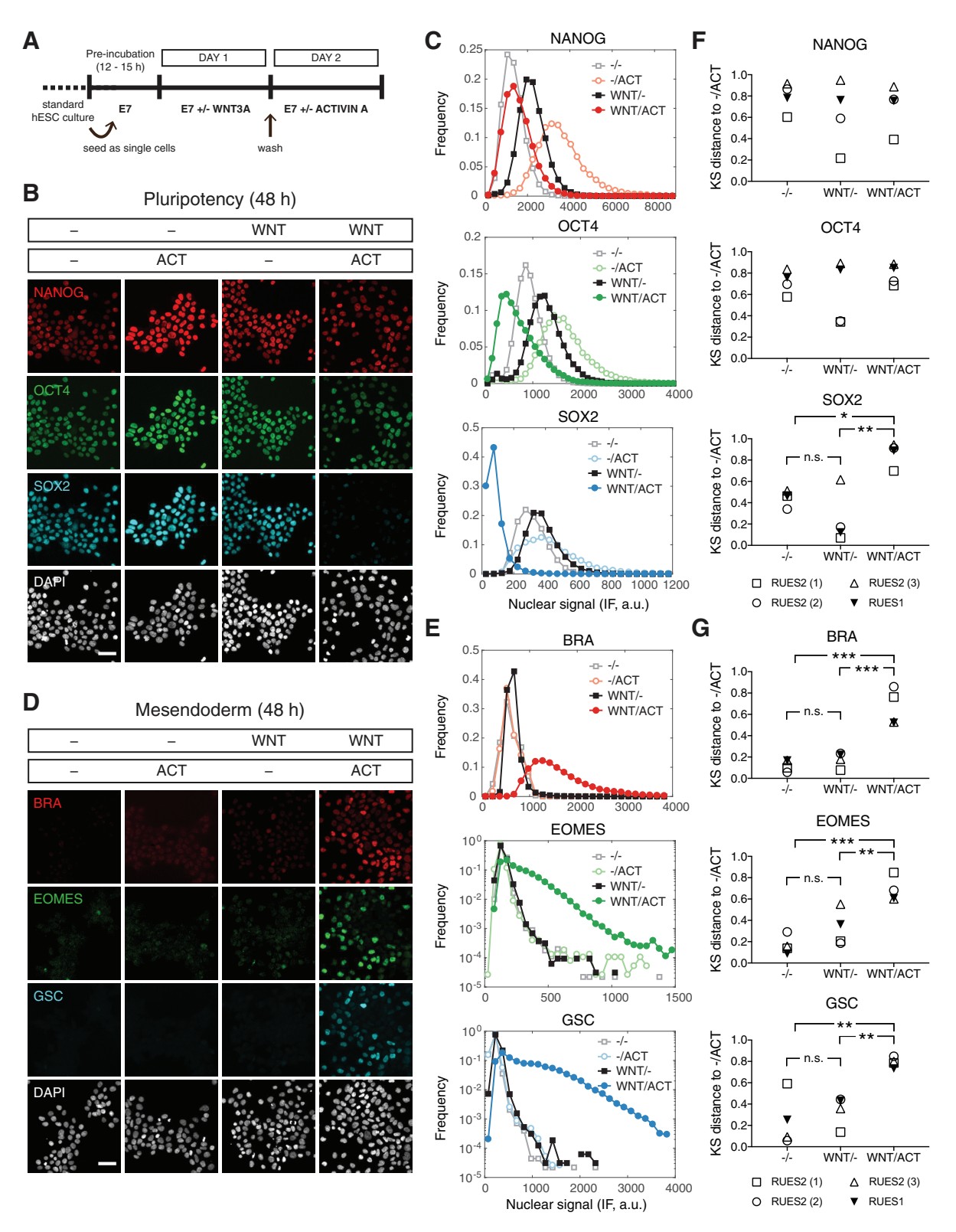

**Figure 6.** WNT priming unveils ACTIVIN-dependent mesendoderm differentiation. (**A**) Schematic outlining the 2 day experimental protocol. (**B–C**) Cells were cultured for one day with or without WNT3A (100 ng/mL, top bar). On the second day they were washed to remove WNT and treated with or without ACTIVIN (10 ng/mL, bottom bar). After the second day cells were fixed and analyzed by immunofluorescence (IF). (**B**) Images: NANOG (red), OCT4 (green), SOX2 (cyan), DAPI (gray). Scale bar, 50 μM. (**C**) Histograms showing the nuclear IF signal quantified in single cells (n > 5,000 cells per

*Figure 6 continued on next page*

*Figure 6 continued*

condition). (**D–E**) Cells were cultured for one day with or without WNT3A (100 ng/mL, top bar). On the second day they were washed to remove WNT and treated with or without ACTIVIN (10 ng/mL, bottom bar). After the second day cells were fixed and analyzed by immunofluorescence (IF). (**D**) Images: BRA (red), EOMES (green), GSC (cyan), DAPI (gray). Scale bar, 50 µM. (**E**) Histograms showing the nuclear IF signal quantified in single cells (n > 5,000 cells per condition). (**F–G**) Kolmogorov-Smirnov (KS) distance of the cumulative probability distribution (CDF) of each marker to the reference CDF (-/ACT) for independent experiments in RUES2 and in RUES1 for the (**F**) pluripotency and (**G**) mesendoderm marker sets. n.s. (or comparison not shown), not significant, *p<0.05, **p<0.01, ***p<0.001, ANOVA.

DOI: https://doi.org/10.7554/eLife.38279.023

The following figure supplement is available for figure 6:

**Figure supplement 1.** ACTIVIN dose-dependent mesendoderm differentiation.

DOI: https://doi.org/10.7554/eLife.38279.024

*Gurdon, 1999*). When presented to isolated animal cap explants that normally give rise only to ectoderm, ACTIVIN was sufficient to induce different types of mesendodermal cells based on its concentration, and was thus qualified in principle as a morphogen (*Green and Smith, 1990*; *Wilson and Melton, 1994*). When the ACTIVIN receptors and the SMAD pathways were characterized in 1990 s, it was shown that micro-injection of different amounts of synthetic mRNAs encoding SMAD2 into *Xenopus* animal caps, also recapitulates the mesendodermal-inducing effects of A ACTIVIN morphogen presentation, independently confirming that the activation of different thresholds of the pathway was sufficient for mesoderm induction (*Shimizu and Gurdon, 1999*). Inhibition of the pathway by microinjection of a dominant negative type II ACTIVIN receptor into *Xenopus* blastula led to complete loss of mesendodermal derivatives, demonstrating that ACTIVIN/SMAD2 signaling was necessary for mesoderm induction in the amphibian embryo (*Hemmati-Brivanlou and Melton, 1992*).

Loss-of-function analysis in the mouse later confirmed the frog conclusions about the pathway as SMAD2/3 double knockout mice failed to properly induce mesendoderm and gastrulate (*Dunn et al., 2004*). However another TGFβ ligand, NODAL, rather than ACTIVIN was shown to be the inducer in the mammalian embryo (*Conlon et al., 1994*). Elegant genetic experiments also performed in the mouse placed NODAL signaling in a positive feedback loop that drives gastrulation. The loop is initiated when NODAL signaling induces BMP4 expression in the extra-embryonic ectoderm. BMP signaling subsequently induces WNT3 expression, which in turn induces high levels of NODAL in the proximal-posterior part of the embryo, which marks the site of primitive streak initiation (*Arnold and Robertson, 2009*).

Using live reporters of SMAD1, SMAD2 and SMAD4, we find that BMP/SMAD1 signaling is stable, ACTIVIN/SMAD2 signaling is transient (or adaptive), and that co-SMAD4 follows the dynamics of the receptor-associated SMADs. In response to ACTIVIN, hESCs transiently induce mesendodermal gene expression, without generating mesendodermal fates, and the cells return to the state of pluripotency. However, the expression of ACTIVIN/NODAL inhibitors remained elevated and plausibly buffer the additional ACTIVIN, as expected from prior theory (*François and Siggia, 2008*). Pre-presentation of WNT, however, stabilizes transcription and fate acquisition without modifying SMAD2 dynamics. Thus, a transient signaling response is compatible with a stable change in cell fate, as shown previously in a murine myoblast cell line (*Sorre et al., 2014*). Both pluripotency maintenance and mesendodermal fate acquisition are not affected by the loss of SMAD3. Our data provide evidence for a previously undetected level of signal integration that implies the presence of a cellular memory. This memory operates not at the level of the SMAD2 signaling dynamics but by some other means to change the response of the cells to ACTIVIN. It is tempting to speculate that it might be mediated through epigenetic mechanisms, in a manner similar to what has been described in *Xenopus* experiments where β-catenin recruits PrMT2 to induce dorsal fates (*Blythe et al., 2010*).

The ACTIVIN/SMAD2 pathway has been shown to be necessary for maintenance of pluripotency, but not sufficient to induce mesendoderm in hESCs, except when influenced by modifiers of other signaling pathways (*D'Amour et al., 2005*; *McLean et al., 2007*; *Singh et al., 2012*; *Funa et al., 2015*). Our study shows that mere pre-treatment of hESCs with WNT, endows them with a memory that enables a graded response to subsequently applied ACTIVIN; simultaneous exposure to WNT and ACTIVIN is not required. Furthermore, the WNT treatment alone is not sufficient to elicit a stable fate change since in minimal media, without ACTIVIN, the cells revert to pluripotency. The ability of embryonic cells to record WNT signals may be a broadly conserved and fundamental aspect of

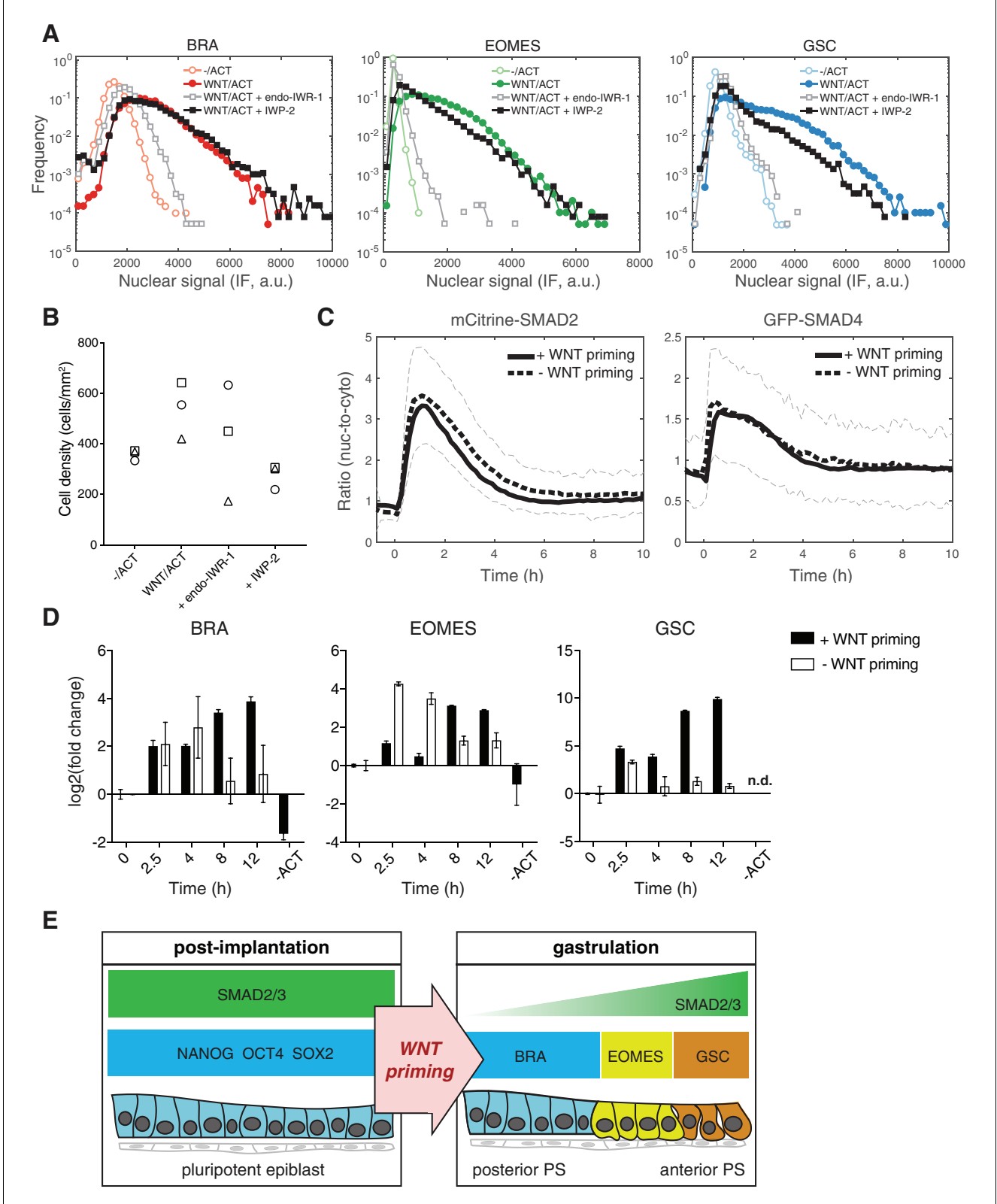

**Figure 7.** WNT priming stabilizes mesendoderm gene transcription without changing SMAD2 and SMAD4 response dynamics. (**A**) Cells were cultured for one day with WNT3A (100 ng/mL). On the second day they were washed to remove WNT and treated with ACTIVIN (10 ng/mL). In order to block WNT signaling through β-catenin cells were treated with endo-IWR-1 (1 μM), which was added with WNT on day one and again with ACTIVIN on day 2. In order to block WNT secretion, cells were similarly treated with IWP-2 (1 μM). After the second day cells were fixed and analyzed by

*Figure 7 continued on next page*

*Figure 7 continued*

immunofluorescence (IF). Histograms show the nuclear IF signal quantified in single cells (n > 5,000 cells per condition). Cells cultured without WNT, corresponding to the pluripotency condition (-/ACT, open circles), represent background signal levels. (B) Quantification of the final cell density in the experiment shown in A and in additional replicates. There is no significant difference between the densities under different conditions (ANOVA). (C) mCitrine-SMAD2 (left) and GFP-SMAD4 (right) response in single cells to ACTIVIN (10 ng/mL) with or without WNT priming (black solid and dashed lines, respectively). ACTIVIN was added at T = 0 hr, which corresponds to 24 hr of WNT stimulation. WNT was removed by washing prior to the addition of ACTIVIN. Images were acquired every 10 min. The black solid and dashed lines represent the average response at each time point and the gray dashed lines represent the population standard deviation of the SMAD2 response without WNT priming (n > 200 cells per time point). Similar results were obtained in two independent experiments. (D) Transcriptional response of mesendoderm genes to ACTIVIN with or without WNT priming. An additional sample was collected that was treated with WNT and left untreated with ACTIVIN (-ACT). Expression in each sample was normalized to GAPDH and then to the level prior to ACTIVIN addition (T = 0 hr). Data represents the mean ±S.D. for n = 3 technical replicates. n.d., not detected. Similar results were obtained in two independent experiments. (E) Model for WNT priming as a requirement for ACTIVIN to function as a morphogen. Following implantation ACTIVIN functions in a dose-independent manner to maintain pluripotency of the epiblast. At the onset of gastrulation, WNT signaling primes cells respond to graded ACTIVIN/SMAD2 signaling, which leads to the anterior-posterior patterning of cells emerging from the primitive streak (PS).

DOI: https://doi.org/10.7554/eLife.38279.025

The following figure supplement is available for figure 7:

**Figure supplement 1.** On-going SMAD2 signaling is required for mesendoderm differentiation but is dispensable for WNT priming.
DOI: https://doi.org/10.7554/eLife.38279.026

animal development, as shown by elegant experiments in *Drosophilia* where the authors demonstrate a morphogen effect for Dpp (a BMP homologue) after cells have lost contact with the source of WNT (*Alexandre et al., 2014*).

In the context of vertebrate development, our findings force the reevaluation of the traditional literature regarding the sufficiency of the ACTIVIN/SMAD2 pathway for mesendodermal induction and patterning. A review of the evidence in model systems for the ACTIVIN/NODAL mediated morphogen effect reveals that in all the experimental settings, at least some of the cells were either still under the influence, or had been previously exposed to WNT signaling before or during ACTIVIN/NODAL signaling. For example, cells of the *Xenopus* animal cap derived from the blastula stage embryo have been under maternal WNT influence as early as the two-cell stage, hours before ACTIVIN is presented to the explants. This influence is the consequence of cortical rotation that occurs after sperm entry, and activates the WNT pathway on the dorsal side of the embryo, as evidenced by the dorsal-specific nuclear localization of β-catenin (*Schneider et al., 1996*; *Larabell et al., 1997*; *Rowning et al., 1997*). It is clear from the asymmetric elongation of the animal cap explants in response to ACTIVIN that the prior WNT exposure sets up a dorsal-ventral pre-pattern that affects the response. It is also clear that the dorsal and ventral regions of the animal cap when separated respond differently to ACTIVIN (*Sokol and Melton, 1991*; *Bolce et al., 1992*). Although some mesoderm is induced in the ventral caps, only cells of the dorsal cap that have seen WNT signals undergo ACTIVIN-mediated induction of GSC + mesendoderm. These experiments demonstrate that both pathways are required for complete mesendodermal patterning without explicitly distinguishing their temporal relationship.

Our study revises this point of view by bringing a temporal order. We suggest that only cells endowed with a WNT memory can respond to ACTIVIN to specify the full range of mesendodermal fates. Even in the extensively studied *Xenopus* embryo, it is still debated whether in the late blastula stage marginal zone, prior to the onset of gastrulation, an ACTIVIN/NODAL gradient already defines the medial to lateral mesendodermal fates or whether there is simply a bipartite division into dorsal organizer and ventral mesoderm (*Harland and Gerhart, 1997*; *Smith, 2009*). This debate also translates into whether ACTIVIN/NODAL is sufficient to pre-pattern the mesendoderm or whether complete patterning requires prior WNT exposure present only in the dorsal part of the embryo. Controversies persist, since the embryo is rapidly developing, there are no live reporters of signaling, and fates are often assayed in early gastrulation rather than in the late blastula. In the mouse the earliest manifestation of the streak is the proximal-posterior expression of WNT, which extends distally. By mid-streak stage, NODAL signaling is highest in the Node (*Tam and Loebel, 2007*). Thus, cells that leave the streak at various proximal-distal positions and NODAL levels, manifestly have been exposed to WNT first (*Tam and Behringer, 1997*). This interpretation, however, does not

eliminate the co-requirement for ACTIVIN and WNT, but rather suggests that instructive WNT, required for mesendodermal differentiation, occurs temporally up-stream of ACTIVIN/SMAD2 signaling.

In conclusion, a cellular memory of WNT signaling, endows embryonic cells, including hESCs, with the competence to maintain ACTIVIN/SMAD2-mediated mesendodermal fate specification and patterning.

# Materials and methods

## Key resources table

| Reagent type (species) or resource | Designation | Source or reference | Identifiers | Additional information |
|---|---|---|---|---|
| Cell line (*Homo sapiens*, XX) | RUES2 | US National Institutes of Health, human ESC registry | human ESC registry no. 0013; RRID:CVCL_VM29 | Human embryonic stem cell line |
| Cell line (*Homo sapiens*, XX) | RUES2-RFP-SMAD1 | this paper | | CRISPR/Cas9-modified human embryonic stem cell line |
| Cell line (*Homo sapiens*, XX) | RUES2-mCit-SMAD2 | this paper | | CRISPR/Cas9-modified human embryonic stem cell line |
| Cell line (*Homo sapiens*, XX) | RUES2-GFP-SMAD4 | PMID: 28760810 | | CRISPR/Cas9-modified human embryonic stem cell line |
| Cell line (*Homo sapiens*, XX) | RUES2-SMAD3-/- clone #1 | this paper | | CRISPR/Cas9-modified human embryonic stem cell line |
| Cell line (*Homo sapiens*, XX) | RUES2-SMAD3-/- clone #2 | this paper | | CRISPR/Cas9-modified human embryonic stem cell line |
| Cell line (*Homo sapiens*, XY) | RUES1 | US National Institutes of Health, human ESC registry | human ESC registry no. 0012; RRID:CVCL_B809 | Human embryonic stem cell line |
| Antibody | anti-Brachyury (goat polyAb) | R and D Systems | Cat. #: AF2085; RRID:AB_2200235 | IF (1:300) |
| Antibody | anti-Brachyury (rabbit mAb) | R and D Systems | Cat. #: MAB20851 | IF (1:200) |
| Antibody | anti-Cdx2 (mouse mAb) | Abcam | Cat. #: ab15258; RRID:AB_2077042 | IF (1:50) |
| Antibody | anti-Eomes (mouse mAb) | R and D Systems | Cat. #: MAB6166; RRID:AB_10919889 | IF (1:200) |
| Antibody | anti-Lamin B1 (rabbit polyAb) | Proteintech | Cat. #: 12987–1-AP; RRID:AB_2136290 | WB (1:2000) |
| Antibody | anti-Goosecoid (goat polyAb) | R and D Systems | Cat. #: AF4086; RRID:AB_2114650 | IF (1:100) |
| Antibody | anti-Nanog (goat polyAb) | R and D Systems | Cat. #: AF1997; RRID:AB_355097 | IF (1:200) |
| Antibody | anti-Oct3/4 (mouse mAb) | BD Biosciences | Cat. #: 611203; RRID:AB_398737 | IF (1:400) |
| Antibody | anti-Smad2 (rabbit mAb) | Cell Signaling | Cat. #: 3122; RRID:AB_10697649 | IF (1:200), WB (1:1000) |
| Antibody | anti-Smad2/3 (mouse mAb) | BD Biosciences | Cat. #: 610842; RRID:AB_398161 | IF (1:100) |
| Antibody | anti-Sox17 (goat polyAb) | R and D Systems | Cat. #: AF1924; RRID:AB_355060 | IF (1:200) |
| Antibody | anti-Sox2 (rabbit mAb) | Cell Signaling | Cat. #: 3579; RRID:AB_2195767 | IF (1:200) |

*Continued on next page*

*Continued*

| Reagent type (species) or resource | Designation | Source or reference | Identifiers | Additional information |
|---|---|---|---|---|
| Peptide, recombinant protein | recombinant human/mouse/rat Activin A | R and D Systems | Cat. #: 338-AC/CF | |
| Peptide, recombinant protein | recombinant human BMP-4 | R and D Systems | Cat. #: 314 BP | |
| Peptide, recombinant protein | recombinant mouse Wnt-3a | R and D Systems | Cat. #: 1324-WN | |
| Peptide, recombinant protein | recombinant human Laminin-521 | BioLamina | | |
| Chemical compound, drug | endo-IWR-1 | Tocris | Cat. #: 3532 | |
| Chemical compound, drug | IWP-2 | Stemgent | Cat. #: 04–0034 | |
| Chemical compound, drug | SB431542 | Stemgent | Cat. #: 04–0010 | |
| Chemical compound, drug | Y-27632 | Abcam | Cat. #: Ab120129 | |
| Software, algorithm | MATLAB | MathWorks | RRID:SCR_001622 | |
| Software, algorithm | StemCellTracker | https://github.com/ChristophKirst/StemCellTracker | | Image segmentation and signal analysis |
| Software, algorithm | R | https://www.r-project.org/ | | |
| Software, algorithm | DESeq2 Bioconductor package | PMID: 25516281 | | Differential gene expression analysis |
| Software, algorithm | PWMEnrich Bioconductor package | https://bioconductor.org/packages/release/bioc/html/PWMEnrich.html | | Motif enrichment analysis |
| Software, algorithm | GOseq Bioconductor package | PMID: 20132535 | | Statistical analysis of over/under represented categories |
| Software, algorithm | AME | PMID: 25953851 | | Motif enrichment analysis |
| Software, algorithm | Tide | PMID: 25300484 | | Sequence trace decomposition |
| Software, algorithm | Cluster 3.0 | PMID: 14871861 | | Clustering |
| Software, algorithm | Java TreeView | PMID: 15180930 | | Data visualization |

## Human embryonic stem cell culture

Experiments were performed with the RUES2 hESC line (XX female; US National Institutes of Health, human ESC registry no. 0013) or CRISPR/Cas9 edited cell lines based on RUES2. Key experiments were repeated with the RUES1 hESC line (XY male; US National Institutes of Health, human ESC registry no. 0012). RUES1 and RUES2 have been authenticated by STR profiling and both tested negative for mycoplasma contamination. hESCs were grown in HUESM medium that was conditioned by mouse embryonic fibroblasts and supplemented with 20 ng/mL bFGF (MEF-CM). Cells were grown on tissue culture dishes (BD Biosciences, San Jose, CA) for maintenance and expansion at 37°C and 5% $CO_2$. Dishes were coated overnight at 4°C with Geltrex (Thermo Fisher Scientific, Waltham, MA) diluted 1:40 in DMEM/F12 and then incubated at 37°C for at least 20 min before passaging. Cells

were passaged as aggregates using Gentle Cell Dissociation Reagent (STEMCELL Technologies, Vancouver, Canada).

## Micropatterned cell culture

Individual micropatterned coverslips (CYTOO, Grenoble, France) were washed one time with water for 5 min at room temperature (RT) to activate the surface according to the manufacturers recommendation. Coverslips were then coated at 37°C for 2 hr with 20 µg/mL Laminin-521 (BioLamina, Sundbyberg, Sweden) in 0.5 mL PBS + Mg/+Ca. The laminin was then removed by serial dilutions without allowing the coverslip to dry (1:4 dilution in PBS –Mg/–Ca, six times). Chips were seeded immediately or stored overnight at 4°C in 2 mL PBS –/– and seeded on the following day.

Cell seeding was performed as follows. Cells growing in MEF-CM were washed once with PBS +/+ and dissociated to single cells with Accutase (STEMCELL Technologies). Cells were centrifuged and 600,000 cells were resuspended in 2 mL of MEF-CM supplemented with 10 µM Rock-inhibitor (Y-27632, Abcam, Cambridge, MA). The cell suspension was then placed over the coverslip in a 35 mm tissue culture dish. The sample was left unperturbed for 10 min at RT in order to achieve homogeneous seeding of the cells throughout the chip before being moved to the incubator. After 2 hr, the medium was replaced with MEF-CM without Rock-inhibitor and the cells were incubated overnight. On the following day, the medium was changed to MEF-CM with additional ligands, BMP4, WNT3A, or ACTIVIN A (R and D Systems, Minneapolis, MN) with or without 10 µM SB431542 (Stemgent, Lexington, MA). We stimulated cells with ACTIVIN A rather than NODAL, because although both ligands differentiate hESCs towards the mesendoderm lineage, NODAL requires much higher concentrations to be effective (*McLean et al., 2007*). For experiments in TeSR-E7 (STEMCELL Technologies), cells were moved to E7 medium the day after seeding and incubated for an additional 24 hr before adding ligands. Live imaging was carried out in E7 imaging medium, which was prepared with FluoroBrite DMEM (Thermo Fisher Scientific) according the published protocol for E8 (*Beers et al., 2012*).

## Single-cell culture

Optical-quality plastic tissue culture dishes (ibidi, Martinsried, Germany) were coated with 10 µg/mL Laminin-521 in PBS +/+ for 2 hr at 37°C or overnight at 4°C. Single cells were collected as described for micropatterned cell culture and dishes were seeded with 50,000 single cells resuspended in 2 mL E7 supplemented with 10 µM Rock-inhibitor. The samples were incubated overnight. On the following day, the medium was changed to E7 supplemented with 10 µM Rock-inhibitor and ligands or small molecules. For the 2 day protocol in which cells were switched from E7 ±100 ng/mL WNT3A to E7 ±10 ng/mL ACTIVIN A, the samples were washed with PBS +/+ before adding fresh medium. Live imaging was carried out in E7 imaging medium. Small molecules were used at the following concentrations and were replaced every 24 hr: 10 µM SB431542 (Stemgent), 1 µM IWP-2 (Stemgent), and 1 µM endo-IWR-1 (Tocris).

## Immunofluorescence and western blotting

Cells on dishes or coverslips were rinsed once with PBS –/–, fixed with 4% paraformaldehyde (Alfa Aesar, Thermo Fisher Scientific, Tewksbury, MA) for 20 min at RT, and then rinsed twice and stored in PBS –/–. Cells were blocked and permeabilized with blocking buffer (2% bovine serum albumin and 0.1% Triton X-100 in PBS –/–) for 30 min at RT. Cells were incubated with primary antibodies in blocking buffer overnight at 4°C and then washed three times with 0.1% Tween-20 in PBS –/– (PBST). Cells were incubated with secondary antibodies (diluted 1:1000): Alexa Fluor 488, 555, or 647-conjugated from (Invitrogen Molecular Probes, Thermo Fisher Scientific) and DAPI nuclear stain (1:10,000 dilution) in blocking buffer for 30 min at RT, and then washed twice with PBST and once with PBS –/–. Dishes were stored and imaged in PBS –/–. Coverslips were mounted on slides using Fluoromount-G mounting medium (SouthernBiotech, Birmingham, AL).

Western blotting was performed using standard techniques. The nuclear fractions presented in *Figure 3—figure supplement 1C* were obtained using the NE-PER Nuclear and Cytoplasmic Extraction Reagents (Thermo Fisher Scientific) and following the manufacturers protocol. Following SDS-PAGE and Western transfer, membranes were stained with anti-Smad2 Rabbit mAb (Cell Signaling Technology, Danvers, MA) at 1:1000 dilution. Membranes were stripped for 15 min at RT using

Restore PLUS Western Blot Stripping Buffer (ThermoFisher Scientific) and re-stained with anti-Lamin B1 Rabbit polyAb (Proteintech, Rosemont, IL) at 1:2000 dilution.

## Imaging

Wide-field images of fixed samples were acquired on an Olympus IX-70 inverted microscope with a 10x/0.4 numerical aperture objective lens. Tiled image acquisition was used to acquire images of large areas of dishes or coverslips in four channels corresponding to DAPI and Alexa Fluor 488, 555, and 647. Live imaging was performed on a spinning disk confocal microscope equipped with 405-, 488-, and 561 nm lasers and an environmental chamber (CellVoyager CV1000, Yokogawa). Images were acquired every 10 min with a 20x/0.75 numerical aperture objective lens. The cells were maintained at 37°C and 5% $CO_2$ during live imaging.

## Generation of SMAD1 and SMAD2 reporter cell lines

For the tagRFP-SMAD1 reporter cells, CRISPR/Cas9 mediated genome engineering was used to fuse a cassette containing a blasticidin resistance gene (BsdR), a T2A self-cleaving peptide, and a tagRFP fluorescent protein onto the N-terminus of SMAD1, so that the locus produces both a tagRFP-SMAD1 fusion protein together with BsdR. Similarly, for the mCitrine-SMAD2 reporter line, CRISPR/Cas9 was used to fuse a cassette containing a puromycin resistance gene (PuroR), a T2A self-cleaving peptide, and an mCitrine fluorescent protein onto the N-terminus of SMAD2, so that the locus produces both an mCitrine-SMAD2 fusion protein together with PuroR.

RUES2 hESCs were nucleofected with a pX335 plasmid (Cong et al., 2013) that co-expresses the nickase version of Cas9 and the specific sgRNA of interest (protospacer sequences: 5'-GCAGCAC TAGTTATACTCCT-3' for SMAD1 and 5'-GGACGACATGTTCTTACCAA-3' for SMAD2), as well as the appropriate homology donor plasmid. Nucleofection was carried out using the Cell Line Nucleofector Kit L (Lonza, Walkersville, MD) and the B-016 setting of a Nucleofector II instrument. Nucleofected cells were plated into MEF-CM supplemented with 10 µM Rock-inhibitor. After 4 days, blasticidin or puromycin was added for 7 days to select for cells that had been targeted. Cells that survived selection were passaged as single cells using Accutase, plated in MEF-CM supplemented with 10 µM Rock-inhibitor, and allowed to grow into colonies. Colonies arising from a single cell were handpicked, expanded, and screened for correct targeting by PCR amplification of the genomic region and Sanger sequencing. Correctly targeted clones were subsequently transfected with ePiggyBac plasmids containing either H2B-mCitrine or H2B-mCherry cassettes to enable nuclear labeling for cell tracking (Lacoste et al., 2009). Individual clones were again isolated and controlled for normal karyotype (G-banding) and pluripotency maintenance.

## Generation of SMAD3 knockout cell lines

RUES2 hESCs were nucleofected with a pX330 plasmid (Cong et al., 2013) that co-expressed the wild-type version of Cas9 and one of two sgRNAs targeting SMAD3 (protospacer sequences: 5'-CCACCAGATGAACCACAGCA-3' for sgRNA #1 and 5'-TTATTATGTGCTGGGGACAT-3' for sgRNA #2). The sgRNAs were designed to target the first or second coding exon shared by all SMAD3 isoforms, a strategy that was successfully used to knockout SMAD3 in human primary cell lines (Voets et al., 2017). Two different sgRNAs were used to control for off-target effects. We modified the pX330 plasmid to also express a puromycin-2A-EGFP cassette to enrich for cells that had been successfully nucleofected. Nucleofection was carried out as described above and cells were plated in MEF-CM supplemented with 10 µM Rock-inhibitor. On the following day puromycin was added for 24 hr. Cells that survived selection were allowed to recover for several days. Cells were then passaged as single cells using Accutase, plated in MEF-CM supplemented with 10 µM Rock-inhibitor, and allowed to grow into colonies. Colonies arising from a single cell were handpicked, expanded, and screened for correct targeting by PCR amplification of the genomic region and Sanger sequencing. The resulting chromatograms were decomposed using the Tide web-based tool (Brinkman et al., 2014).

## RT-PCR

RUES2 cells were seeded in 6-well plates (1,00,000 single cells per well) in E7 medium supplemented with 10 µM Rock-inhibitor and incubated overnight. On the following day the medium was changed

to E7 supplemented with 10 µM Rock-inhibitor with or without 10 ng/mL ACTIVIN A. Samples (three pooled-wells) were collected in 1 mL Trizol at 0, 2.5, 4, 6 and 12 hr. The 2 day protocol was carried out similarly: the medium was changed on the day after seeding to E7 supplemented with 10 µM Rock-inhibitor with or without 100 ng/mL WNT3A. On the second day following seeding the wells were washed once with PBS and the media was changed to E7 supplemented with 10 µM Rock-inhibitor with or without 10 ng/mL ACTIVIN A. Samples were collected in 1 mL Trizol before the addition of WNT3A, after WNT3A treatment (referred to as 0 hr), and at 2.5, 4, 8, and 12 hr after the addition of ACTIVIN A. Total cellular RNA for each sample was extracted using the RNeasy Mini Kit (QIAGEN, Germantown, MD) and cDNA was synthesized using the Transcriptor First Strand cDNA Synthesis Kit (Roche). RT-PCR for selected genes was performed on three technical replicates using the LightCycler 480 SYBR Green I Master mix in a LightCycler 480 instrument (Roche, Basel, Switzerland). Primers were designed using Primer-BLAST (*Ye et al., 2012*) or obtained from qPrimer-Depot (*Cui et al., 2007*) or from previously published sequences (*Mendjan et al., 2014*). Primer sequences and source are listed in *Table 1*.

## Total RNA-sequencing

RUES2 cells were seeded in 6-well plates (200,000 single cells per well) in E7 medium supplemented with 10 µM Rock-inhibitor and incubated overnight. On the following day the media was changed to E7 supplemented with 10 µM Rock-inhibitor with or without 10 ng/mL ACTIVIN A. Samples (three pooled-wells) were collected in 1 mL Trizol at 1, 2.5, 4, 8 and 12 hr for the ACTIVIN-treated conditions and after 0, 6 and 12 hr for the no-ACTIVIN conditions (to be used as negative controls). Total cellular RNA for each sample was extracted using RNeasy Mini Kit (QIAGEN) and 2 ug of total RNA was used to prepare each individual RNA-seq library. RNA-seq library construction was conducted with the TruSeq RNA Library Preparation Kit (Illumina, San Diego, CA) as per the manufacturer's instructions and sequenced in an Illumina HiSeq 2500 apparatus. Raw reads were mapped to hg19 using STAR aligner, and the gene read counts were normalized using the DESeq2 Bioconductor package (*Love et al., 2014*). Library preparation, sequencing, and mapping were performed by the New York Genome Center (New York, NY, USA). All raw data files are available from the GEO database (accession number GSE111717).

## Image analysis

For images acquired from micropatterned cell culture experiments, stitching and colony detection were carried out as described previously using custom software written in MATLAB (*Etoc et al., 2016*). For analysis of the SMAD reporter lines on micropatterned colonies a single z-plane through the middle of the colony was analyzed at each time point. Background in each channel was removed by subtracting a minimum intensity image that was generated by taking the minimum value at each pixel over all images in that channel in the same z-plane. Vignetting was corrected by dividing by a

**Table 1.** RT-PCR primer sequences and source.

| Gene symbol | Forward primer | Reverse primer | Source |
|---|---|---|---|
| NANOG | TCCAACATCCTGAACCTCAGC | ACCATTGCTATTCTTCGGCCA | Primer-BLAST |
| OCT4 | AAACCCACACTGCAGCAGAT | TGTGCATAGTCGCTGCTTGA | Primer-BLAST |
| SOX2 | TACAGCATGATGCAGGACCA | CCGTTCATGTAGGTCTGCGA | Primer-BLAST |
| NODAL | AGACATCATCCGCAGCCTAC | CAAAAGCAAACGTCCAGTTCT | Primer-BLAST |
| GAPDH | AATCCCATCACCATCTTCCA | TGGACTCCACGACGTACTCA | Primer-BLAST |
| SMAD7 | CCAGGCTCCAGAAGAAGTTG | CCAACTGCAGACTGTCCAGA | qPrimerDepot |
| LEFTY1 | CTCCATGCCGAACACCAG | GGAAAGAGGTTCAGCCAGAG | qPrimerDepot |
| LEFTY2 | TCAATGTACATCTCCTGGCG | CTGGACCTCAGGGACTATGG | qPrimerDepot |
| BRA | CGTTGCTCACAGACCACAG | ATGACAATTGGTCCAGCCTT | qPrimerDepot |
| GSC | GAGGAGAAAGTGGAGGTCTGGTT | CTCTGATGAGGACCGCTTCTG | Mendjan et al. |
| EOMES | CACATTGTAGTGGGCAGTGG | CGCCACCAAACTGAGATGAT | Mendjan et al. |

DOI: https://doi.org/10.7554/eLife.38279.027

flat-field image that was generated by normalizing the intensity of the minimum image to 1. This procedure corrects the intensity drop-off at the border of the images without further altering the average image intensity.

Nuclei segmentation and signal quantification were performed on the corrected images as follows. The H2B image was thresholded to generate a binary image separating the foreground (nuclei) from the background. The original, corrected H2B image was then filtered with a median and sphere filter with parameters matching the expected size of individual nuclei. Local maxima corresponding to individual nuclei were detected using the MATLAB extended-maxima transform function. Maxima were dilated to increase the likelihood of obtaining a single maximum per nuclei. Maxima falling within the foreground were used as seeds for watershed segmentation, which was also restricted to the foreground and was used to obtain a labeled object corresponding to each nucleus within the image. The segmented objects were further processed to eliminate objects much larger or smaller than the expected size of individual nuclei. The results of the segmentation were used as a mask to obtain the median per cell nuclear intensity in each channel. Images acquired from fixed samples that were stained by immunofluorescence were similarly segmented and analyzed. The DAPI channel was used to perform nuclei segmentation, and the segmented objects were subsequently used to obtain the median per cell signal intensity in each channel.

For the mCitrine-SMAD2 reporter line an enrichment of the mCitrine signal in the cytoplasm relative to the nucleus prior to ligand presentation could be detected, which decreased concomitantly with an increase in the nuclear signal following ACTIVIN presentation. Therefore, the mCitrine-SMAD2 response was quantified as the nuclear-to-cytoplasmic ratio, which was used previously as a readout for TGFβ pathway activity (*Warmflash et al., 2012*). In order to estimate the cytoplasmic signal for each cell, a narrow donut surrounding the nuclear mask was formed by dilating the nuclear mask once by an inner radius and a second time by an outer radius and subtracting the first dilated object from the second. The donut, which formed the cytoplasmic mask, was not restricted to the foreground pixels (H2B signal), but it was prevented from overlapping with the masks of neighboring nuclei. The median mCitrine intensity within the nuclear mask was divided by the median intensity within the cytoplasmic mask on a per cell basis. The qualitative behavior of the SMAD2 nuclear-to-cytoplasmic response was not sensitive to the exact size of the inner and outer radius of the donut. Therefore, the values were chosen manually and kept fixed throughout all analyses. For the RFP-SMAD1 reporter line a faint nuclear signal could be detected prior to ligand presentation. However, no cytoplasmic signal could be detected above the background. Therefore, the RFP-SMAD1 response was quantified as the median nuclear signal normalized to the median H2B signal to normalize for cells moving in and out of the z-plane. In order to analyze the SMAD response as a function of radial position within the micropatterned colonies, the cells within a single colony were binned based on the radial position of their center and the average response per cell within each bin was calculated. The radial profile for individual colonies was then averaged over several colonies. Analysis of the SMAD reporter lines in single-cell culture was carried out in a similar manner. The maximum intensity projection image, rather than a single z-plane was analyzed. Since cells remained flat under these conditions it was not necessary to normalize the SMAD1 signal by H2B.

## RNAseq analysis and clustering

For each gene, a baseline expression profile, which was calculated using a linear interpolation between the 0, 6 and 12 hr control samples, was subtracted from the expression values of the ACTIVIN-treated samples. The gene list was filtered to contain only those genes that: (1) had at least one time point with an absolute fold-change larger than 2 (up- or down-regulated) compared to 0 hr and (2) had at least one time point with a normalized read count higher than 100. That generated a list of 3529 genes of interest, which was then hierarchically clustered by their z-scored expression values, using Cluster 3.0 (*de Hoon et al., 2004*) with the following options: centered correlation as the similarity metric and average linkage as clustering method. The resulting hierarchical tree was visualized using Java TreeView (*Saldanha, 2004*) to identify the minimal clusters of interest.

## Motif analysis

To identify motifs enriched within gene clusters, the 2000 bp upstream sequences for all genes were extracted using the PWMEnrich Bioconductor package. Motif enrichment of each cluster's sequence

set was performed using AME (*Bailey et al., 2015*) with the HOCOMOCOv10 database (*Kulakovskiy et al., 2016*) against the background (upstream sequences of all genes).

## Gene enrichment analysis

We assessed the enrichment of the genes in each of the groups identified in *Figure 4A* for previously defined marker gene sets from isolated endoderm, mesoderm, and ectoderm/epiblast tissue from E7.5 mouse embryos (*Lu et al., 2018*) using the GOseq Bioconductor package (*Young et al., 2010*). Mouse genes were mapped to human orthologues using data downloaded from the Mouse Genome Informatics database (http://www.informatics.jax.org/downloads/reports/HOM_MouseHumanSequence.rpt).

## Data and software availability

The RNA-sequencing data related to *Figure 4* and discussed in this publication have been deposited in NCBI's Gene Expression Omnibus (*Edgar et al., 2002*) and are accessible through GEO Series accession number GSE111717 (https://www.ncbi.nlm.nih.gov/geo/query/acc.cgi?acc=GSE111717). Image analysis software StemCellTracker (commit no. 1c31fdd) is available through GitHub (*Kirst, 2014*; copy archived at https://github.com/elifesciences-publications/StemCellTracker).

# Acknowledgements

We are thankful to all the members of the Brivanlou and Siggia groups for helpful discussions and to C Zierhut for critical reading of the manuscript. Our research was supported by NIH grants R01 HD080699 and R01 GM101653 to AHB and EDS and NSF grant DGE 132526 to AY. Imaging was performed at The Rockefeller University Bio-Imaging Resource Center.

# Additional information

### Funding

| Funder | Grant reference number | Author |
| --- | --- | --- |
| National Science Foundation | Graduate Research Fellowship DGE132526 | Anna Yoney |
| National Institute of General Medical Sciences | Research Project R01GM101653 | Eric D Siggia Ali H Brivanlou |
| Eunice Kennedy Shriver National Institute of Child Health and Human Development | Research Project R01HD080699 | Eric D Siggia Ali H Brivanlou |

The funders had no role in study design, data collection and interpretation, or the decision to submit the work for publication.

### Author contributions

Anna Yoney, Conceptualization, Formal analysis, Investigation, Visualization, Methodology, Writing—original draft, Writing—review and editing, Gave final approval of the version to be published; Fred Etoc, Conceptualization, Investigation, Writing—original draft, Writing—review and editing, Gave final approval of the version to be published; Albert Ruzo, Thomas Carroll, Resources, Formal analysis, Visualization, Writing—review and editing, Gave final approval of the version to be published; Jakob J Metzger, Formal analysis, Writing—review and editing, Gave final approval of the version to be published; Iain Martyn, Shu Li, Investigation, Writing—review and editing, Gave final approval of the version to be published; Christoph Kirst, Software, Writing—review and editing, Gave final approval of the version to be published; Eric D Siggia, Ali H Brivanlou, Conceptualization, Supervision, Funding acquisition, Writing—original draft, Writing—review and editing, Gave final approval of the version to be published

## Author ORCIDs

Anna Yoney (iD) https://orcid.org/0000-0003-4988-9237
Christoph Kirst (iD) https://orcid.org/0000-0003-4867-5288
Eric D Siggia (iD) https://orcid.org/0000-0001-7482-1854
Ali H Brivanlou (iD) http://orcid.org/0000-0002-1761-280X

## Decision letter and Author response

Decision letter https://doi.org/10.7554/eLife.38279.032
Author response https://doi.org/10.7554/eLife.38279.033

## Additional files

### Supplementary files

• Transparent reporting form
DOI: https://doi.org/10.7554/eLife.38279.028

### Data availability

Sequencing data have been deposited in GEO under accession code GSE111717.

The following dataset was generated:

| Author(s) | Year | Dataset title | Dataset URL | Database and Identifier |
|---|---|---|---|---|
| Yoney A, Etoc F, Ruzo A, Carroll T, Metzger JJ, Martyn I, Li S, Kirst C, Siggia ED, Brivanlou AH | 2018 | WNT signaling memory is required for ACTIVIN to function as a morphogen | https://www.ncbi.nlm.nih.gov/geo/query/acc.cgi?acc=GSE111717 | NCBI Gene Expression Omnibus, GSE111717 |

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
