## [Decision Letter]

Thank you for submitting your article "WNT signaling memory is required for ACTIVIN to function as a morphogen in human gastruloids" for consideration by *eLife*. Your article has been reviewed by three peer reviewers, one of whom is a member of our Board of Reviewing Editors, and the evaluation has been overseen by Aviv Regev as the Senior Editor. The reviewers have opted to remain anonymous.

The reviewers have discussed the reviews with one another and the Reviewing Editor has drafted this decision to help you prepare a revised submission.

The work described in this manuscript is based on the manipulation of a very useful in vitro human "gastruloid" system that the Brivanlou and Siggis labs have developed. In this case, experiments are directed at clarifying the interactions between WNT and ACTIVIN signaling that serve to regulate the transition between pluripotency and differentiation. The main conclusion is that a relatively brief exposure to a WNT ligand modifies pluripotent cells such that they respond by differentiating when subsequently treated with ACTIVIN under conditions normally insufficient to trigger differentiation. That is, the "memory" of WNT signaling changes the response to ACTIVIN.

General Assessment:

The reviewers were uniformly enthusiastic about the quality of the work in the manuscript, as well as its potential for clarifying some aspects of signaling during early development, particularly early human development. The reviewers' comments fall into two general categories. The first requires some additional experiments to clarify some details of the quantification applied to your imaging results and the use of small molecule inhibitors. The second asks for some revisions in the text and, perhaps, some changes in the manner in which certain of your conclusions are stated.

Required revisions requiring additional work:

Use of reagents and quantification:

1) Can the authors clarify the effects of small molecules singly and in combination on the total number of cells to determine if there were any effects related to toxicity?

2) Figure 2D, G; Figure 3C, E, G: Isn't the relevant metric the number of cells with positive nuclear stain?

3) Subsection “Single-cell response dynamics reflect the behavior at the edge of gastruloids”: Can the authors clarify how they define SMAD2 baseline and, therefore, how they determine the percent of cells above baseline?

4) Subsection “WNT memory stabilizes the mesendodermal transcriptional response to ACTIVIN”, first paragraph and Figure 7: What is the evidence that IWP-2 continues to block WNT secretion for 48 hrs?

Other aspects of data interpretation/metrics:

1) The conclusion that ACTIVIN alone does not induce mesoderm relies on just three markers (BRA, EOMES and GSC) and the persistence of some markers associated with pluripotency (yet most do not seem very specific to a pluripotent state, they resemble more of an ACTIVIN response). Is there additional evidence for lack of mesoderm induction by ACTIVIN alone (perhaps in the RNA-seq data)? The lack of induction is at odds with the ability of ACTIVIN, NODAL or NODAL/Vg1 alone to induce non-organizer mesoderm in the absence of a β-catenin signal in amphibian (animal cap) and mammalian cells. The authors discuss this important point but is the apparent discrepancy between systems due to the inability to separate endogenous WNT from ACTIVIN/NODAL/Vg signals or possibly an evolutionary difference between mammalian and non-mammalian systems? How similar is mesendoderm induced by WNT+ACTIVIN to "natural" mammalian (mouse) or even nonmammalian mesendoderm? It seems gene expression data associated with Figure 4 could be compared to published datasets or previous characterizations.

2) Can the authors detail how many biological replicates underpin each major experiment?

Revisions that do not require additional experiments:

1) Why was ACTIVIN used as the agent to induce SMAD2 signaling in the experiments? Is not NODAL(s) the more relevant ligand "in vivo"? Certainly real human embryos are off limits to providing a direct answer, but in hESCs are endogenous ACTIVIN or NODAL genes induced by mesendoderm-initiating WNT (or BMP) treatment? ACTIVIN and NODAL ligands are often used interchangeably to affect SMAD2/3 signaling in vertebrate embryonic cells or ESCs, but these different ligands can produce different effects, so a brief rationale for using ACTIVIN and not NODAL would be welcome.

2) The authors should temper "conclusions" that go beyond the data presented. The authors tend to rule out SMAD3 engagement in the ACTIVIN response perhaps over-weighting the SMAD3 mutant studies. However, is there anything to rule out a simple redundancy between SMAD2/3 where either contributes? Insight from SMAD2 and SMAD2/3 mutant hESCs which are presumably on hand would help to settle this. The authors state "Overall we conclude that cells maintain a memory of WNT exposure which makes them competent to differentiate in response to ACTIVIN (Figure 7E)." this seems quite reasonable. However, the authors go on to write in the last sentence of the Results "The molecular mechanism of this memory occurs through β-catenin-induced transcriptional changes as a modifier of SMAD2 signaling dynamics". There is nothing to directly back this up: β-catenin was never studied, WNT transcriptional actions were not directly addressed, and the sentence is best deleted and possible mechanisms with supporting evidence left for discussion.

---

## [Author Response]

Required revisions requiring additional work:Use of reagents and quantification:1) Can the authors clarify the effects of small molecules singly and in combination on the total number of cells to determine if there were any effects related to toxicity?

No toxicity effect was observed on the total number of cells with any of the small molecules, SB (10 µM), endo-IWR1 (1 µM), or IWP2 (1 µM). Quantification of the total cell number in the experiments using small molecules has been added to Figure 5—figure supplement 1D and Figure 7B.

2) Figure 2D, G; Figure 3C, E, G: Isn't the relevant metric the number of cells with positive nuclear stain?

We think that the average nuclear or nuclear-to-cytoplasmic intensity is the correct metric. To determine the number of positive nuclei requires a binary threshold, which discards data about the levels of the SMAD response that we can obtain from our live reporters. In all of our datasets, with the exception of SMAD1 on micropatterned colonies, the SMAD response is not binary and single cells respond with dynamics that are similar to the average response. This can be seen as a shift in the distribution (or histogram) of the response across all cells. We should have presented this data in our original manuscript. We have now added it in Figure 3—figure supplement 1. In the experiments in which we followed the SMAD1 response in micropatterned colonies, we were able to determine an on/off threshold. Therefore, following the reviewer’s suggestion, we also quantified the number of SMAD1 positive cells and found results that are qualitatively similar to those presented in Figure 2D in which we present the average nuclear intensity as a function of time and position within the colony. This additional quantification is also now shown in Figure 2—figure supplement 2C–D.

3) Subsection “Single-cell response dynamics reflect the behavior at the edge of gastruloids”: Can the authors clarify how they define SMAD2 baseline and, therefore, how they determine the percent of cells above baseline?

We should have been more precise in our definition of the baseline. The baseline is defined by the average SMAD2 nuclear-to-cytoplasmic ratio at T > 8 h after ACTIVIN presentation. This definition was previously included in Figure 3—figure supplement 1 and it has now been added to the Results subs “Single-cell response dynamics reflect the behavior at the edge of gastruloids”.

4) Subsection “WNT memory stabilizes the mesendodermal transcriptional response to ACTIVIN”, first paragraph and Figure 7: What is the evidence that IWP-2 continues to block WNT secretion for 48 hrs?

The original paper reporting IWP2 (Chen et al., 2009) demonstrated complete inhibition of WNT by 24 hours of compound exposure in mammalian cells. In hESCs specifically (Blauwkamp et al., 2012), presenting IWP-2 and replacing it every 24 hours was also shown to completely inhibit WNT. We also replaced the IWP-2 every 24 hours, following previously published protocols, and this has been clarified in the Results subsection “WNT memory stabilizes the mesendodermal transcriptional response to ACTIVIN”, and in the Materials and methods section.

In order to verify that the IWP-2 was functioning to block endogenous WNT at 1 µM, we tested it in a side-by-side experiment, in which micropatterned colonies were treated with BMP4 +/- IWP-2 (1 µM). In this context, mesendoderm is dependent on endogenous WNT production and signaling (Martyn et al., 2018). We find that mesendoderm is blocked by 1 µM IWP-2 in BMP4-treated micropatterned colonies. This data is presented in Figure 7—figure supplement 1B–C.

Other aspects of data interpretation/metrics:1) The conclusion that ACTIVIN alone does not induce mesoderm relies on just three markers (BRA, EOMES and GSC) and the persistence of some markers associated with pluripotency (yet most do not seem very specific to a pluripotent state, they resemble more of an ACTIVIN response). Is there additional evidence for lack of mesoderm induction by ACTIVIN alone (perhaps in the RNA-seq data)? The lack of induction is at odds with the ability of ACTIVIN, NODAL or NODAL/Vg1 alone to induce non-organizer mesoderm in the absence of a β-catenin signal in amphibian (animal cap) and mammalian cells. The authors discuss this important point but is the apparent discrepancy between systems due to the inability to separate endogenous WNT from ACTIVIN/NODAL/Vg signals or possibly an evolutionary difference between mammalian and non-mammalian systems? How similar is mesendoderm induced by WNT+ACTIVIN to "natural" mammalian (mouse) or even nonmammalian mesendoderm. It seems gene expression data associated with Figure 4 could be compared to published datasets or previous characterizations.

Following the reviewer’s advice, to further rule out mesoderm induction in our ACTIVIN only stimulation, we have also verified that canonical mammalian mesodermal markers such as HAND1, EVX1, MESP1, MESP2, MSGN, and TBX6 are not induced (Figure 4—source data 2). We take the oddity of lack of mesoderm induction by ACTIVIN alone in our system wholeheartedly, as we appreciate the importance of evaluating the relevance of our synthetic in vitro platform in modeling “natural” embryos. As the reviewer(s) has noted we address this important point in our Discussion section. We argue that while we cannot eliminate possible evolutionary differences in the fine-tuning of mesendodermal induction, we believe that our findings are in support of the classical experimental embryological discoveries made in the amphibian system. We state that “Although some mesoderm is induced in the ventral animal cap explants of *Xenopus*, only cells of the dorsal caps that have seen Wnt signals undergoes ACTIVIN-mediated induction of GSC+ mesendoderm”. This is in agreement with our “cellular memory” of WNT signaling in our system.

Following the reviewer’s suggestion, we additionally compared our RNA-seq dataset presented in Figure 4 with the tissue specific RNA-seq datasets of mouse embryos at E7.5 (Lu et al., 2018). This comparison showed that from the 3 groups defined in Figure 4, group 1 (transient induction following ACTIVIN presentation) displays the largest representation of endodermal genes, with 129 genes, out of 2,956 total in group 1, overlapping with the mouse endoderm gene set. This data along with the fact that GSC is induced under the WNT/ACTIVIN condition (RT-PCR and IF) suggests our mesendoderm is of dorso-anterior character. Group 2 (genes that are stably induced following ACTIVIN presentation) showed some enrichment of both endodermal (27) and mesodermal (15) genes, out of 452 total in group 2. Group 3 (genes that are down-regulated following ACTIVIN presentation) matched most closely primitive ectoderm/epiblast (5 genes), out of 121 total in group 3. This information has now been added to Figure 4—figure supplement 1B and the lists of overlapping genes have been added as Figure 4—source data 5.

2) Can the authors detail how many biological replicates underpin each major experiment?

Two or more biological replicates were performed for each major experiment. This information was added to the figure legends.

Since the absolute intensity measurements from immunofluorescence can vary between experiments performed on different days, we only directly compared samples processed at the same time and these are presented as the nuclear intensity histograms shown on a single plot. In order to facilitate comparison across biological replicates processed and imaged on different days, we calculated the Kolmogorov-Smirnov distance between the cumulative distribution functions of each condition to a defined reference condition as described in the last paragraph of the subsection “Long-term, elevated SMAD2 baseline maintains pluripotency”. The cumulative distribution function is calculated from the fluorescence intensity histogram. This type of analysis is robust to changes in imaging parameters, as long as the parameters are fixed between different conditions and their corresponding reference condition. Statistical tests were performed on KS distance estimates.

Revisions that do not require additional experiments:1) Why was ACTIVIN used as the agent to induce SMAD2 signaling in the experiments? Is not NODAL(s) the more relevant ligand "in vivo"? Certainly real human embryos are off limits to providing a direct answer, but in hESCs are endogenous ACTIVIN or NODAL genes induced by mesendoderm-initiating WNT (or BMP) treatment? ACTIVIN and NODAL ligands are often used interchangeably to affect SMAD2/3 signaling in vertebrate embryonic cells or ESCs, but these different ligands can produce different effects, so a brief rationale for using ACTIVIN and not NODAL would be welcome.

The reviewer(s) is correct: the more relevant ligand “in vivo” in the mouse embryo is NODAL. Additionally, we have shown in Etoc et al. (2016) that BMP4 induces NODAL, in Martyn et al. (2018) that WNT induces NODAL, and in our present study that ACTIVIN also induces NODAL in a stable manner. We and others choose to use ACTIVIN rather than NODAL, because although both ligands differentiate hESCs towards the mesendoderm lineage, NODAL requires much higher concentrations to be effective (McLean et al., 2007). Following, the reviewer’s recommendation we have now included this rational in the last paragraph of the Materials and methods subsection “Micropatterned cell culture”.

2) The authors should temper "conclusions" that go beyond the data presented. The authors tend to rule out SMAD3 engagement in the ACTIVIN response perhaps over-weighting the SMAD3 mutant studies. However, is there anything to rule out a simple redundancy between SMAD2/3 where either contributes? Insight from SMAD2 and SMAD2/3 mutant hESCs which are presumably on hand would help to settle this. The authors state "Overall we conclude that cells maintain a memory of WNT exposure which makes them competent to differentiate in response to ACTIVIN (Figure 7E)." this seems quite reasonable. However, the authors go on to write in the last sentence of the Results "The molecular mechanism of this memory occurs through β-catenin-induced transcriptional changes as a modifier of SMAD2 signaling dynamics". There is nothing to directly back this up: β-catenin was never studied, WNT transcriptional actions were not directly addressed, and the sentence is best deleted and possible mechanisms with supporting evidence left for discussion.

We are grateful to the reviewer(s) for pointing out two overreaching statements. First, it is correct that while our RUES2-SMAD3^-/-^ mutant does not change the response to ACTIVIN, we cannot rule out possible redundancy between SMAD2 and 3. We have, therefore, included this possibility in our interpretation in the subsection “SMAD3 is dispensable for the response of the adaptive and stable gene classes to ACTIVIN”. Second, while we show that endo-IWR-1, which blocks WNT signaling at the level of β-catenin, does eliminate ACTIVIN-mediated mesendoderm induction, we did not provide *direct* evidence that the molecular mechanism underlying the memory occurs through β-catenin-induced transcriptional changes. Thus, we removed the sentence quoted by the reviewer from the text.